# Pharmacological CDK4/6 inhibition promotes vulnerability to lysosomotropic agents in breast cancer

Jamil Nehme [1], Sjors Maassen[1], Sara Bravaccini[2], Michele Zanoni [2], Caterina Gianni[2], Ugo De Giorgi[2], Abel Soto-Gamez [1], Abdullah Altulea[1], Teodora Gheorghe[1], Boshi Wang [1]✉ & Marco Demaria [1]✉

## Abstract

Breast cancer is a leading cause of mortality worldwide. Pharmacological inhibitors of cyclin-dependent kinases (CDK) 4 and 6 (CDK4/6i) inhibit breast cancer growth by inducing a senescent-like state. However, the long-term treatment efficacy remains limited by the development of drug resistance, so clearance of senescent-like cancer cells may extend the durability of treatment. However, we show here that while CDK4/6i-treated breast cancer cells exhibit various senescence-associated phenotypes, they remain insensitive to common senolytic compounds. By searching for novel vulnerabilities, we identify a significantly increased lysosomal mass and altered lysosomal structure across various breast cancer cell types upon exposure to CDK4/6i in preclinical systems and clinical specimens. We demonstrate that these CDK4/6i-induced lysosomal alterations render breast cancer cells sensitive to lysosomotropic agents, such as L-leucyl-L-leucine methyl ester (LLOMe) and salinomycin. Importantly, sequential treatment with CDK4/6i and lysosomotropic agents effectively reduces the growth of both hormone receptor-positive (HR⁺) and subsets of triple-negative breast cancer (TNBC) cells in vivo. This sequential therapeutic strategy offers a promising approach to eliminate CDK4/6i-induced senescent(-like) cells, potentially reducing tumor recurrence and enhancing the overall efficacy of breast cancer therapy.

**Keywords** CDK4/6 Inhibitors; Abemaciclib; Cellular Senescence; p16; Lysosome
**Subject Categories** Cancer; Cell Cycle; Organelles

## Introduction

Despite considerable improvements in early detection and treatment, cancer remains one of the leading causes of death worldwide. Breast cancer is the most prevalent type of cancer among women

(Wilkinson and Gathani, 2022). One of the most common anticancer strategies relies on targeting mechanisms that lead to unrestrained proliferation. This can be accomplished by inflicting high amounts of DNA damage using genotoxic stressors such as chemotherapeutic agents or ionizing radiation. However, many chemotherapeutics used to treat cancer lack specificity and their systemic administration is associated with multiple short- and long-term adverse reactions (Demaria et al, 2017; Yao et al, 2020). More targeted antiproliferative approaches, such as inhibiting Cyclin-dependent Kinases (CDK) 4 and 6 to block the cell cycle transition from the G1 to S phase, have been shown to be better tolerated and have significantly fewer side effects (Wang et al, 2022). The CDK4/6 inhibitors (CDK4/6i) palbociclib, ribociclib, and abemaciclib have been approved by the FDA for the treatment of metastatic hormone receptor-positive (HR⁺) and human epidermal growth factor receptor 2-negative (HER2⁻) breast cancer (Eggersmann et al, 2019). These agents have demonstrated to provide a notable improvement in progression-free survival (PFS) compared with endocrine therapy alone (Maltoni et al, 2024; O'Leary et al, 2016; Rocca et al, 2014; Rocca et al, 2017). However, CDK4/6i are rarely cytotoxic but more often cytostatic. Our laboratory has previously shown that treatment with CDK4/6i causes senescence-like phenotypes associated with transient or permanent growth arrest, depending on the genetic background and treatment duration (Wang and Demaria, 2021; Wang et al, 2022). Cell cycle arrest is one of the hallmarks of cellular senescence. Therefore, the promotion of cancer cell senescence is a potent barrier to tumorigenesis and the desired outcome of cancer treatment (Ewald et al, 2010; Lee and Schmitt, 2019). While cancer senescence can be induced at a lower drug dose than that required for cell death, thus reducing potential toxicity, there is evidence that subpopulations of senescent cancer cells may re-enter the cell cycle, acquire stem cell-like phenotypes, and promote disease recurrence (Evangelou et al, 2023; Schmitt et al, 2022). Therefore, a second therapy that can remove senescent-like cancer cells by targeting the vulnerability induced by the first drug, a strategy known as the one-two punch, is a promising approach for maximizing therapeutic efficacy (Sieben et al, 2018; Wang and Bernards, 2018).

Lysosomes are crucial for supporting the enhanced proliferative capacity of cancer cells because of their roles in cellular metabolism,

[1]European Research Institute for the Biology of Ageing (ERIBA), University Medical Center Groningen (UMCG), University of Groningen (RUG), Groningen, The Netherlands. [2]IRCCS Istituto Romagnolo per lo Studio dei Tumori (IRST) "Dino Amadori", Meldola, Italy. ✉E-mail: b.wang01@umcg.nl; m.demaria@umcg.nl

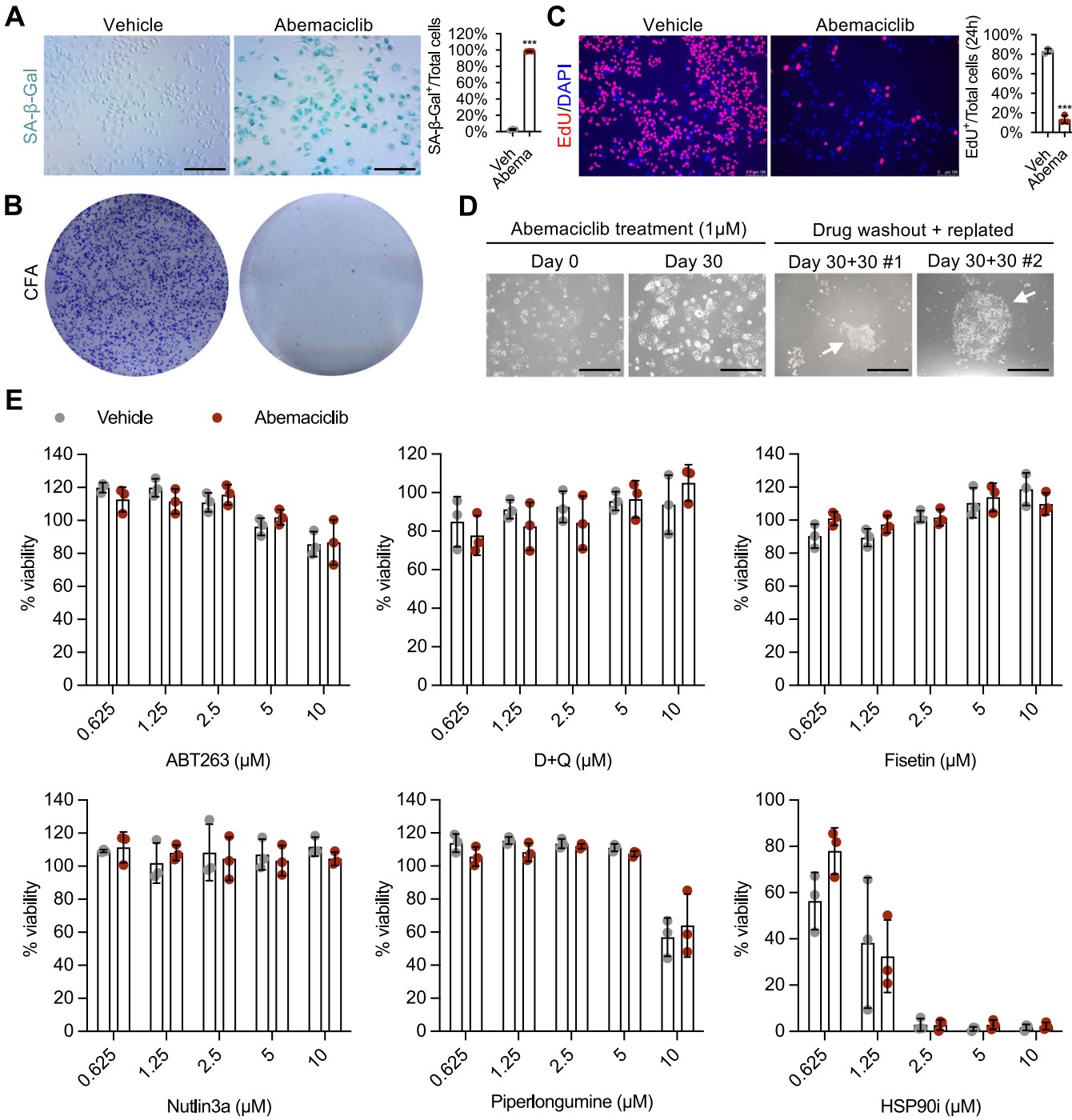

**Figure 1. Abemaciclib-induced senescent MCF-7 cells are insensitive to common senolytic compounds.**

(A, B) MCF-7 cells were treated with either vehicle (water) or abemaciclib (1 µM for 8 days), then cells were replated for SA-β-Gal staining (scale bar 1 mm) and quantified (A) or colony formation assay (for 8 days of culture) (B), $n = 3$ independent experiments, $p < 0.0001$. (C) MCF-7 cells were treated with vehicle (water) or abemaciclib (1 µM for 8 days), replated, incubated with EdU (10 µM) for 20 h, stained with EdU/DAPI (scale bar, 100 µm), and quantified. $n = 3$ independent experiments; $p < 0.0001$. (D) MCF-7 cells were treated with abemaciclib (1 µM) for 30 days (refreshed every 48 h), the drug was then washed out and cells were replated as single cells and cultured in drug-free media for an additional 30 days. White arrows indicate colonies formed by a single 'escaped' cell (scale bar, 1 mm). (E) MCF-7 cells were treated with either vehicle (water) or abemaciclib (1 µM for 8 days), and then treated with common senolytic compounds. Cell viability was measured using MTS assay ($n = 3$ independent experiments). For (A, C), unpaired Student's $t$-tests (two-tailed) were used. For (E), two-way ANOVA. Data are shown as the mean ± SD. ***$p < 0.001$. Source data are available online for this figure.

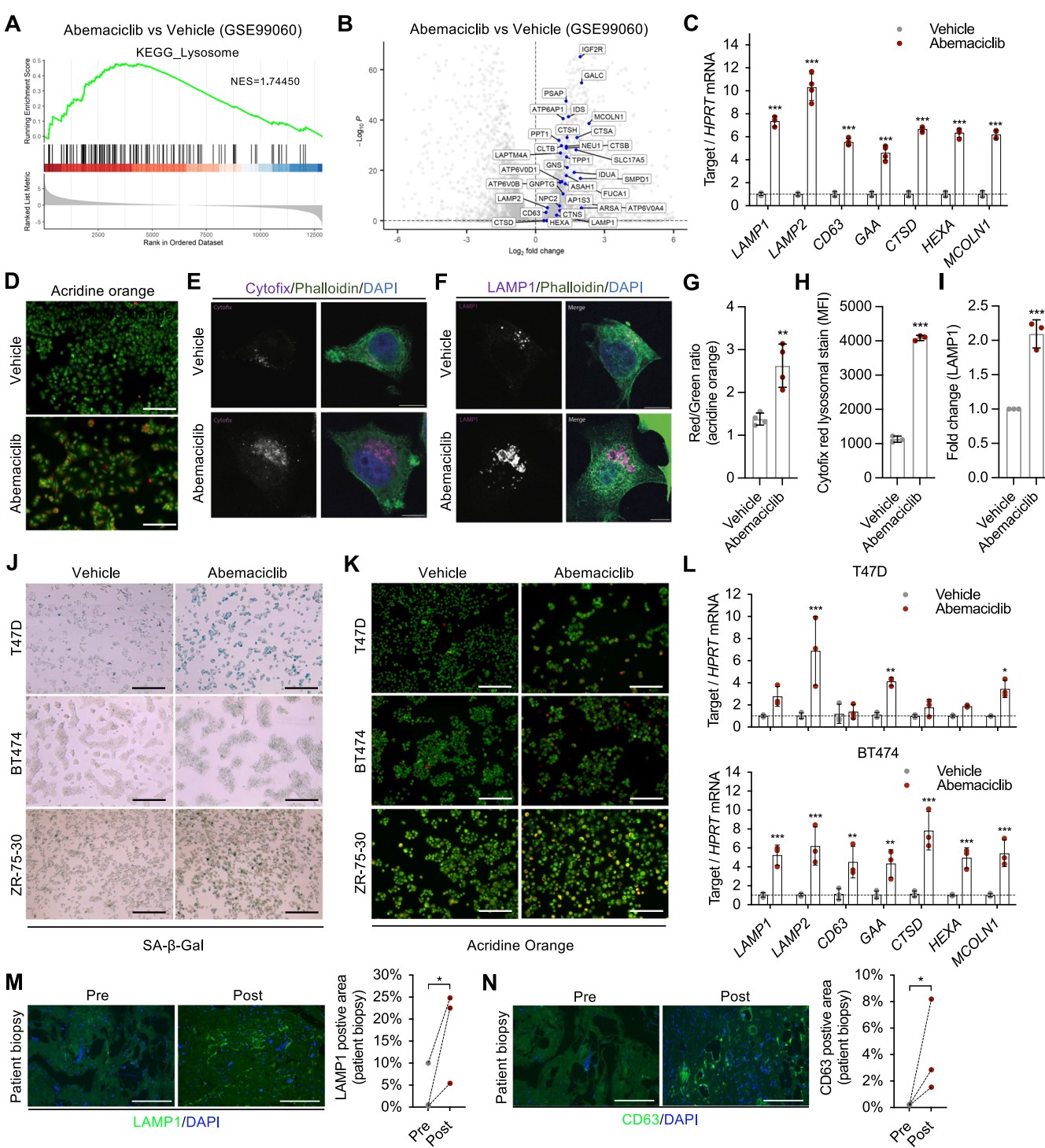

© The Author(s)

The EMBO Journal  Volume 44 | Issue 7 | April 2025 | 1921–1942    **1923**

nutrient recycling, and regulation of cellular processes vital for survival and growth (Kallunki et al, 2013). Cancer cells often exhibit alterations in their lysosomal content, size, and structure (Kallunki et al, 2013). These changes render cancer cells vulnerable to lysosomal membrane permeabilization (LMP). Agents that induce LMP, also known as lysosomotropic agents, have emerged as novel therapeutic strategies for the treatment of different types of cancer (Hu and Carraway, 2020).

Changes in lysosomal content and structure are also a major senescence-associated phenotype, often used for the identification of senescent cells (Hernandez-Segura et al, 2018).

In this study, we observed lysosomal changes as a prominent phenotype induced by CDK4/6i in different breast cancer cell types and evaluated the potential therapeutic effect of exposing CDK4/6i-treated breast cancer cells to lysosomotropic agents.

**Figure 2. Abemaciclib increases lysosomal mass in HR+ breast cancer cells.**

(A, B) GSEA plot showing the enrichment for the KEGG_lysosome pathway genes (A) and a volcano plot highlighting lysosomal genes that were upregulated (B) in abemaciclib-treated MCF-7 cells compared to vehicle control. The expression data were obtained from GSE99060. $n = 3$ samples/group. (C–I) MCF-7 cells were treated with vehicle (water) or abemaciclib (1 μM for 8 days) and then replated for qRT-PCR targeting lysosomal genes ($P < 0.001$ for all genes) (C), acridine orange staining (scale bar, 1 mm) (D), cytofix staining (scale bar, 10 μm) (E), and LAMP1 staining (scale bar, 10 μm) (F). Staining for acridine orange ($p = 0.0031$) (G), cytofix ($p < 0.0001$) (H), and LAMP1 ($p = 0.0007$)(I) was quantified by flow cytometry analyses, $n = 3$ or 4 independent experiments. (J–L) T47D, BT474 and ZR-75-30 cells were treated with vehicle (water) or abemaciclib (1 μM for 8 days), then replated for SA-β-Gal staining (scale bar, 1 mm) (J), acridine orange staining (scale bar, 1 mm) (K), and qRT-PCR of lysosomal genes (For T47D, $p_{LAMP2} < 0.001$, $p_{GAA} = 0.004$, $p_{MCOLN1} = 0.02$; For BT474, $p_{CD63} = 0.002$, $p_{GAA} = 0.002$, other genes $p < 0.001$) (L), $n = 3$ independent experiments. (M, N) tumor biopsies from patients pre- and post-treatment with CDK4/6 inhibitors were stained for LAMP1 ($p = 0.048$) (M) or CD63 ($p = 0.0119$) (N) (scale bar, 250 μm), quantified, and compared (post vs. pre), $n = 3$ patients. For (B), Wald test (deseq2 package). For (C, L), two-way ANOVA was performed. For (G–I), unpaired Student's t-tests (two-tailed) were used. For (M, N), a paired Student's t-test (one-tailed) was used. Data are shown as the mean ± SD. *$p < 0.05$, **$p < 0.01$, ***$p < 0.001$. Source data are available online for this figure.

# Results

## Abemaciclib induces a senescent phenotype unresponsive to typical senolytics in breast cancer cells

To assess whether exposure to CDK4/6i confers senescence-like features to HR+ breast cancer cells, we treated MCF-7 cells with abemaciclib for 8 days. Treatment led to enlarged and flattened morphology, increased senescence-associated β-galactosidase (SA-β-Gal) activity, and diminished clonogenic capacity and EdU incorporation, indicating a senescent state (Fig. 1A–C). Similar to what we have previously reported in the context of normal cell (Wang et al, 2022), treatment of MCF-7 cells with abemaciclib was accompanied by upregulation of p53, but not NF-κB, target genes (Fig. EV1A). However, even after prolonged exposure to abemaciclib (30 days), signs of senescence escape were observed upon drug withdrawal, as evidenced by rescued proliferative activity (Fig. 1D). This observation was in line with previous cell culture and in vivo reports suggesting that senescent cancer cells might escape a stable cell cycle arrest (Evangelou et al, 2023), and supports the idea that a second sequential treatment is necessary to increase therapeutic efficacy. Considering the appearance of senescence-like features, we examined the sensitivity of abemaciclib-treated cells to various common senolytic compounds. However, abemaciclib-treated cells were insensitive to all tested senolytics (Fig. 1E). Notably, most, if not all, senolytic compounds induce cell death via the apoptotic pathway. However, cancer cells often develop mutations that render them resistant to apoptosis (Kulbay et al, 2022). Thus, the identification of drugs capable of triggering cell death through mechanisms independent of the apoptotic pathway could represent a more effective strategy.

## Abemaciclib induces lysosomal mass expansion in HR+ breast cancer

To uncover potential vulnerabilities induced by abemaciclib treatment, we analyzed the gene expression profiles of treated and untreated MCF-7 cells using public RNA-seq data (Goel et al, 2017), in which a notable downregulation of cell cycle genes and upregulation of the senescence signature SenMayo were observed (Fig. EV1B,C). Among the top differentially regulated genes, lysosomal genes and genes associated with lysosomal pathways and functions were significantly upregulated following abemaciclib treatment at various concentrations and durations (Goel et al, 2017; Hafner et al, 2019; Watt et al, 2021) (Figs. 2A,B and EV2A–C).

Elevated expression of lysosomal genes was validated by qPCR using an independent experimental sample set (Fig. 2C). Transcription factor TFEB is a master regulator of lysosomal biogenesis. Interestingly, previous studies have demonstrated that CDK4/6 can phosphorylate TFEB, inducing its translocation from the nucleus to the cytoplasm and thereby inactivating its transcriptional function (Yin et al, 2020). In accordance, we observed that upon abemaciclib treatment TFEB was retained in the nucleus (Fig EV2D). We then evaluated how the altered expression of lysosomal genes impacted lysosomal content and structure. Acridine orange staining demonstrated an increase in the quantity of red fluorescent acidic vesicles, indicative of elevated lysosomal mass (Fig. 2D). This observation was further supported by elevated CytoFix™ Red lysosomal staining, indicative of increased lysosomal mass (Fig. 2E). Immunostaining revealed elevated expression of the lysosomal structural protein LAMP1, highlighting an increased number of lysosomes and changes in lysosomal size (Figs. 2F and EV2E). We further confirmed the increased fluorescence intensity of acridine orange, CytoFix™ Red, and LAMP1 in cells exposed to abemaciclib using flow cytometry (Fig. 2G–I). Importantly, changes in lysosomal activity (SA-β-Gal), mass (acridine orange), and gene expression were observed in multiple HR+ breast cancer cell lines after abemaciclib treatment (Figs. 2J–L and EV2F). Finally, we examined whether lysosomal alterations could be detected in breast cancer patients treated with CDK4/6i. To accomplish this, we performed LAMP1 and CD63 immunostaining on tumor biopsies collected from the same patients before and after treatment. Strikingly, immunostaining for the lysosomal proteins LAMP1 and CD63 significantly increased after exposure to CDK4/6i in all patients (Fig. 2M,N). Taken together, these data suggest that treatment with CDK4/6i causes consistent and reproducible lysosomal alterations in human cancer cells and biopsies.

## Abemaciclib increases HR+ breast cancer cell sensitivity to lysosomotropic agents-induced cell death

An increase in lysosomal biogenesis along with changes in lysosomal size and structure could potentially render abemaciclib-treated cancer cells vulnerable to lysosomotropic agent-induced cell death. First, we used L-leucyl-L-leucine methyl ester (LLOMe), a lysosomotropic detergent assembled by lysosomal enzymes as a condensation product (Thiele and Lipsky, 1990). MCF-7 cells were exposed to abemaciclib (or vehicle) for 8 days, followed by treatment with LLOMe for 48 h. The viability of cells exposed to sequential abemaclicb/LLOMe treatment was severely

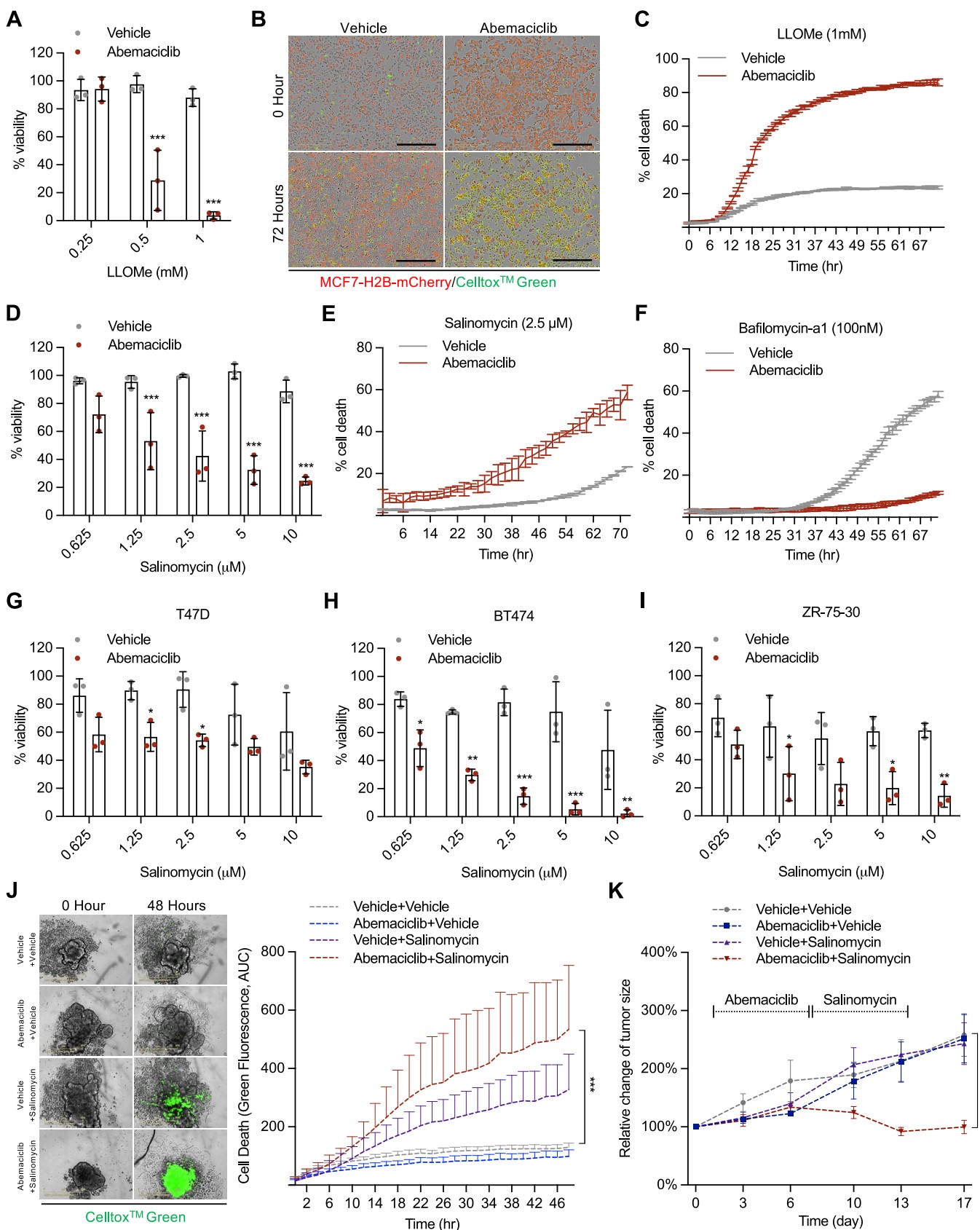

**Figure 3.  Abemaciclib sensitizes HR+ breast cancer cells to lysosomotropic agent-induced cell death.**

(A–F) MCF-7 cells (labeled with H2b-mCherry) were treated with vehicle (water) or abemaciclib (1 µM for 8 days) and then re-plated for subsequent treatment with LLOMe. Cell viability was measured by MTS assay ($p < 0.0001$) (A), and cell death was assessed using IncuCyte live cell imaging with the death marker Celltox™ Green, scale bar, 400 µm (B) for representative images and (C) for quantification. For subsequent treatments with salinomycin, cell viability was measured using MTS assay ($p_{1.25µM} = 0.0005$, other concentrations $p < 0.0001$) (D), while cell death was monitored using IncuCyte imaging (E). After treatment with bafilomycin A1, cell death was measured using IncuCyte imaging (F). $n = 3$ independent experiments. (G–I) Cells from T47D ($p_{1.25µM} = 0.0412$, $p_{2.5µM} = 0.022$) (G), BT474 ($p_{0.625µM} = 0.0151$, $p_{1.25µM} = 0.0017$, $p_{10µM} = 0.0015$, other concentrations $p < 0.0001$) (H), and ZR-75-30 ($p_{1.25µM} = 0.0456$, $p_{5µM} = 0.0124$, $p_{10µM} = 0.0037$) (I) lines were treated with either vehicle (water) or abemaciclib (1 µM for 8 days), then replated for further treatment with salinomyin. Cell viability was measured using the MTS assay ($n = 3$ independent experiments). (J) MCF-7 spheroids were formed and treated with vehicle or abemaciclib (1 µM for 6 days), followed by vehicle or salinomycin (5 µM for 2 days) treatment. Cell death was measured using IncuCyte imaging with Celltox™ Green, and the results were quantified and analyzed ($n = 4$ spheroids/group, $P < 0.0001$). (K) Foxn1$^{Nu}$ mice bearing MCF-7 tumors were treated with abemaciclib (40 mg/kg for 7 days) or salinomycin (5 mg/kg for 7 days) or both (abemaciclib/salinomycin sequential treatments). Tumor volume was measured over time, and the tumor growth curve was plotted ($n = 4$ or 6 mice/group, $p < 0.0001$). For (A, C–I), data are shown as mean ± SD, two-way ANOVA. For (J, K), data are shown as the mean ± SEM, two-way ANOVA. *$p < 0.05$, **$p < 0.01$, ***$p < 0.001$. Source data are available online for this figure.

affected (Fig. 3A). To monitor cell death in real time, we transduced MCF-7 cells with an mCherry-tagged histone 2b (H2b) reporter and incubated the cells with a green fluorescent indicator (Celltox™ Green) of plasma membrane integrity loss. Interestingly, abemaciclib-treated cells exhibited high sensitivity to cell death upon exposure to LLOMe, as measured by the high number of double-positive cells relative to vehicle-treated cells (Fig. 3B,C). Only a partial rescue of cell viability was obtained when we incubated Abemaciclib and LLOMe-treated cells with the pan-caspase inhibitor Q-VD-OPh (QVD), suggesting that caspase activation is not the sole inducer of cell death in this context (Fig. EV2G). Loss-of-function mutations in the oncosuppressor retinoblastoma protein 1 (RB1) are associated with increased resistance to CDK4/6 inhibitors (Antonarelli et al, 2023). Interestingly, analysis of an RNAseq dataset of MCF-7 cells carrying a short hairpin against RB1 and treated with abemaciclib revealed decreased expression of lysosomal genes (Fig. EV2C). We observed a similar downregulation of lysosomal genes in abemaciclib-treated MCF-7 cells carrying siRNAs against RB1, which also correlated with a reduced sensitivity to LLOMe (Fig. EV2H,I). These data suggest that RB1 plays a role in modulating enhanced lysosomal biogenesis and sensitivity to lysosomotropic agents in cells treated with abemaciclib. Although LLOMe is a well-known and potent lysosomal detergent, its application in translational settings remains limited (Kavcic et al, 2020). Therefore, we tested salinomycin, a compound that has recently been shown to accumulate iron in lysosomes, consequently increasing ROS and leading to LMP (Mai et al, 2017), as well as bafilomycin A1, which inhibits vacuolar H+-ATPase (V-ATPase), causing lysosomal deacidification. Salinomycin effectively induced cell death in the abemaciclib-treated cells (Fig. 3D,E). In contrast, no effect was observed with bafilomycin A1 (Fig. 3F), suggesting that the abemaciclib-treated cells were insensitive to lysosomal deacidification. Based on these data, we selected salinomycin for further investigation. Similar to what was observed for MCF-7 cells, HR+ cancer cell lines BT474, T47D, and ZR-75-30 showed increased sensitivity to salinomycin-induced cell death following abemaciclib (Fig. 3G–I). To determine whether sequential treatment with abemaciclib/salinomycin affects the viability of normal cells, we initially evaluated its effect in BJ and IMR90 fibroblasts. However, normal cells tolerated the treatment well, suggesting selective toxicity in cancer cells (Fig. EV3A). To further validate this selectivity, we performed co-culture experiments with fluorophore-

labeled MCF-7 cells and BJ fibroblasts. Sequential treatment with abemaciclib/salinomycin induced the death of MCF-7 cells without affecting the viability of normal cells (Fig. EV3B,C).

Next, we evaluated whether sequential treatment could induce cancer cell toxicity in clinically relevant 3D structures. MCF-7 cells were grown in spheroids and exposed to abemaciclib and salinomycin, either alone or in combination. As observed in 2D cells, sequential treatment significantly increased the number of dead cells (Fig. 3J). Finally, we tested the efficacy of the sequential treatment in vivo. MCF-7 cells were injected into the flanks of nude (Foxn1$^{Nu}$) mice. The mice were then divided into different groups and treated with vehicle, abemaciclib, salinomycin, or a sequential combination of the two drugs. Compared with single treatment, the sequential treatment effectively restrained tumor growth and decreased tumor size (Fig. 3K). Altogether, these data suggest that sequential treatment with abemaciclib and lysosomotropic agents selectively achieves toxicity in HR+ breast cancer cells both ex vivo and in vivo.

## Abemaciclib induces a senescence-like phenotype in TNBC and enhances susceptibility to lysosomotropic agents

CDK4/6i, including abemaciclib, have received FDA approval for the treatment of advanced HR+, HER2− metastatic breast cancer but are also under investigation in multiple preclinical and clinical studies for the treatment of other solid tumors, including triple-negative breast cancer (TNBC) (Hu et al, 2021). Most TNBC cells undergo transient growth arrest during treatment with CDK4/6i in cell culture, and proliferation is quickly restored after drug withdrawal (Goel et al, 2017; Wang and Demaria, 2021). Accordingly, we observed that abemaciclib induced cell cycle arrest in the TNBC cell line MDA-MB-231 during treatment, with cells resuming proliferation upon drug withdrawal (Fig. EV4A,B). Despite the lack of stable cell cycle arrest, abemaciclib-treated MDA-MB-231 cells showed alterations in lysosomal function, structure, and morphology, as exemplified by high SA-β-Gal activity, elevated expression of numerous lysosomal genes, and increased acridine orange and LAMP1 staining, reflecting altered lysosomal content and size (Figs. 4A–F and EV4C,D). In contrast, abemaciclib failed to promote lysosomal alterations in another TNBC cell line, BT549, which was shown to be resistant to CDK4/6i treatment (O'Brien et al, 2018). Interestingly, when we used a

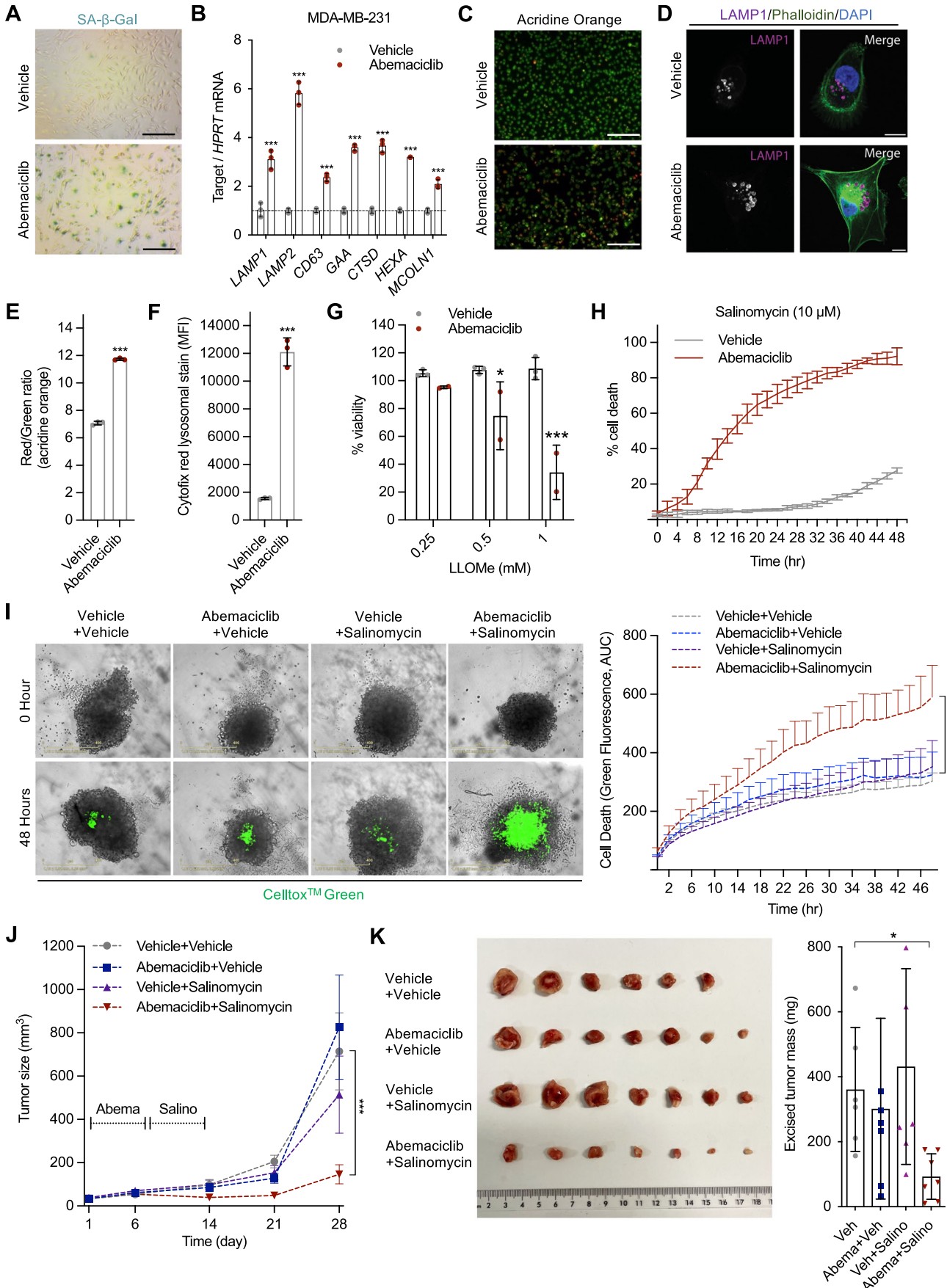

Figure 4. Abemaciclib sensitizes triple-negative breast cancer to lysosomotropic agent-induced cell death.

(A–F) MDA-MB-231 cells were treated with either vehicle (water) or abemaciclib (1 μM for 8 days). After treatment, the cells were subjected to SA-β-Gal staining (scale bar, 1 mm) (A), qRT-PCR of lysosomal genes ($p < 0.001$ for all genes) (B), acridine orange staining (scale bar, 1 mm) (C), and LAMP1 staining (scale bar, 10 μm) (D). The intensities of acridine orange ($p < 0.0001$) (E) and cytofix ($p < 0.0001$) (F) staining were quantitatively analyzed using flow cytometry. $n = 3$ or 4 independent experiments. (G, H) Vehicle-or abemaciclib (1 μM for 8 days)-treated MDA-MB-231 cells were subsequently treated with LLOMe, and cell viability was quantified using MTS assay ($p_{0.5mM} = 0.0315$, $p_{1mM} = 0.0001$) (G), or with salinomycin and their death was quantified using IncuCyte imaging with Celltox™ Green (H), $n = 3$ independent experiments. (I) MDA-MB-231 spheroids were treated with either vehicle or abemaciclib (1 μM for 6 days), followed by vehicle or salinomycin (5 μM for 2 days). Cell death was measured using IncuCyte imaging with Celltox™ Green and quantification was plotted ($n = 4$ spheroids/group, $p < 0.0001$). (J, K) Foxn1$^{Nu}$ mice bearing MDA-MB-231 tumors were treated with abemaciclib (40 mg/kg for 7 days) or salinomycin (5 mg/kg for 7 days) or both (abemaciclib/salinomycin sequential treatments), tumor volume was measured over time and data were used to plot tumor growth curve, $p < 0.0001$ (J), excised tumors were weighed, and weights were plotted, $p = 0.0476$ (K), $n = 6$ or 7 mice/group. For (B, G, H), data are shown as mean ± SD, two-way ANOVA. For (E, F), data are shown as mean ± SD, unpaired Student's $t$-test (two-tailed). For (I, J), data are shown as the mean ± SEM, two-way ANOVA. For (K), data are shown as the mean ± SD, one-way ANOVA. *$p < 0.05$, **$p < 0.01$, ***$p < 0.001$. Source data are available online for this figure.

fluorescent probe to track CDK4/6 enzymatic activity, we observed high activity even in the presence of abemaciclib, suggesting alterations in the lysosomal biogenesis are consequence of the on-target effect of abemaciclib (Fig. EV4E–I) Next, we evaluated whether abemaciclib rendered MDA-MB-231 cells more sensitive to the toxic effect of lysosomotropic agents. Similar to what was observed in HR$^+$ cell lines, LLOMe, and salinomycin, but not bafilomycin A1, showed significant toxicity in abemaciclib-treated MDA-MB-231 cells (Figs. 4G,H and EV4J), whereas no effect was observed in BT549 cells (Fig. EV4K). Sequential treatment with abemaciclib and salinomycin effectively induced cell death in MDA-MB-231 3D spheroid culture (Fig. 4I). Finally, we tested the sequential treatment in vivo. MDA-MB-231 cells were inoculated into the mammary fat pads of nude mice. Mice were then exposed to vehicle, abemaciclib, and salinomycin, either alone or in combination. Only for the sequential treatment, a significantly reduced tumor size during the treatment phase and the absence of substantial tumor growth after treatment withdrawal were observed (Fig. 4J,K). These data suggest that sequential therapy abemaciclib/lysosomotropic agents might be applied to triple-negative breast cancer, provided that abemaciclib inhibits CDK4/6 activity.

### Lysosomal enlargement enhances sensitivity to lysosomotropic agents-induced cell death

To assess the efficacy of other CDK4/6i in sensitizing breast cancer cells to lysosomotropic agents, MCF-7 cells were treated with palbociclib instead of abemaciclib. Interestingly, while palbociclib increased the sensitivity to both LLOMe and salinomycin, it did so to a lesser extent than abemaciclib (Fig. 5A). Previous reports have demonstrated that abemaciclib, unlike other CDK4/6i inhibitors, induces intracellular vacuoles originating from acidic vesicles, including late endosomes and lysosomes (Hino et al, 2020). We observed vacuoles in cells treated with abemaciclib, but not with palbociclib (Fig. 5B). Consequently, abemaciclib-treated cells had larger lysosomes than palbociclib-treated cells, as revealed by LAMP1 staining (Fig. 5C).

We then tested whether forcing the formation of vacuoles was sufficient to sensitize the cells to lysosomotropic agent-induced cell death. Vacuolin-1 is a cell-permeable compound that inhibits lysosomal fission and significantly increases the size of endo/lysosomal compartments, leading to vacuolization (Choy et al, 2018; Sano et al, 2016). Treatment of MCF-7 and MDA-MB-231 cells with vacuolin-1 resulted in a substantial increase in vacuole formation, but did not trigger a senescent phenotype (Figs. 5D,E and EV5A, B). Importantly, both MCF-7 and MDA-MB-231 cells

exposed to vacuolin-1 were highly sensitive to LLOMe and salinomycin (Fig. 5F,G). Finally, we evaluated whether vacuolin-1 could promote lysosomal alterations and vulnerability to lysosomotropic agents in cells insensitive to abemaciclib-induced lysosomal perturbations. Remarkably, the treatment of BT549 cells with vacuolin-1 resulted in pronounced intracellular vacuolization (Fig. 5H) and enhanced sensitivity to lysosomotropic agent-induced cell death (Fig. 5I). These findings indicate that the observed differences in sensitivity can be related, at least in part, to differences in lysosomal vacuolization, thereby extending the potential application of an alternative treatment approach to cancer cells that exhibit a limited response to CDK4/6i treatment.

## Discussion

CDK4/6 inhibitors, particularly abemaciclib, have emerged as promising agents for the treatment of HR$^+$ and HER2$^-$ breast cancers. However, several limitations of their efficacy remain, necessitating the development of improved therapeutic approaches. Prolonged inhibition of CDK4/6 activity in normal cells can promote various senescence-associated features, including durable and stable cell cycle arrest (Wang et al, 2022). Although cell cycle arrest is an intrinsic tumor suppressor mechanism, other senescence-associated phenotypic changes, including resistance to apoptosis and stemness, can contribute to breast cancer survival and dissemination (Evangelou et al, 2023). Thus, senescent-like cancer cells that can escape cell cycle arrest due to acquired genetic and epigenetic aberrations may have increased aggressiveness (McGrath et al, 2024). Therefore, sequential therapy based on a senescence-like phenotype inducer followed by a compound that takes advantage of the vulnerability developed by the initial treatment might be an efficient way to selectively eliminate tumor cells and reduce the risk of recurrence (Sieben et al, 2018). In this study, we showed that common senolytic compounds, most of which promote apoptosis, failed to eliminate breast cancer cells treated with abemaciclib. A particularly conserved phenotype across various models of senescence is the altered lysosomal activity and mass. Various studies have highlighted the significance of lysosomal biogenesis in the development of the senescence phenotype as well as its role in conferring resistance to different treatments (Curnock et al, 2023; Fassl et al, 2020; Li et al, 2023; Llanos et al, 2019; Martinez-Carreres et al, 2019; Rovira et al, 2022). Accordingly, we observed profound lysosomal changes in both HR$^+$ and triple-negative breast cancer cells, which were independent of

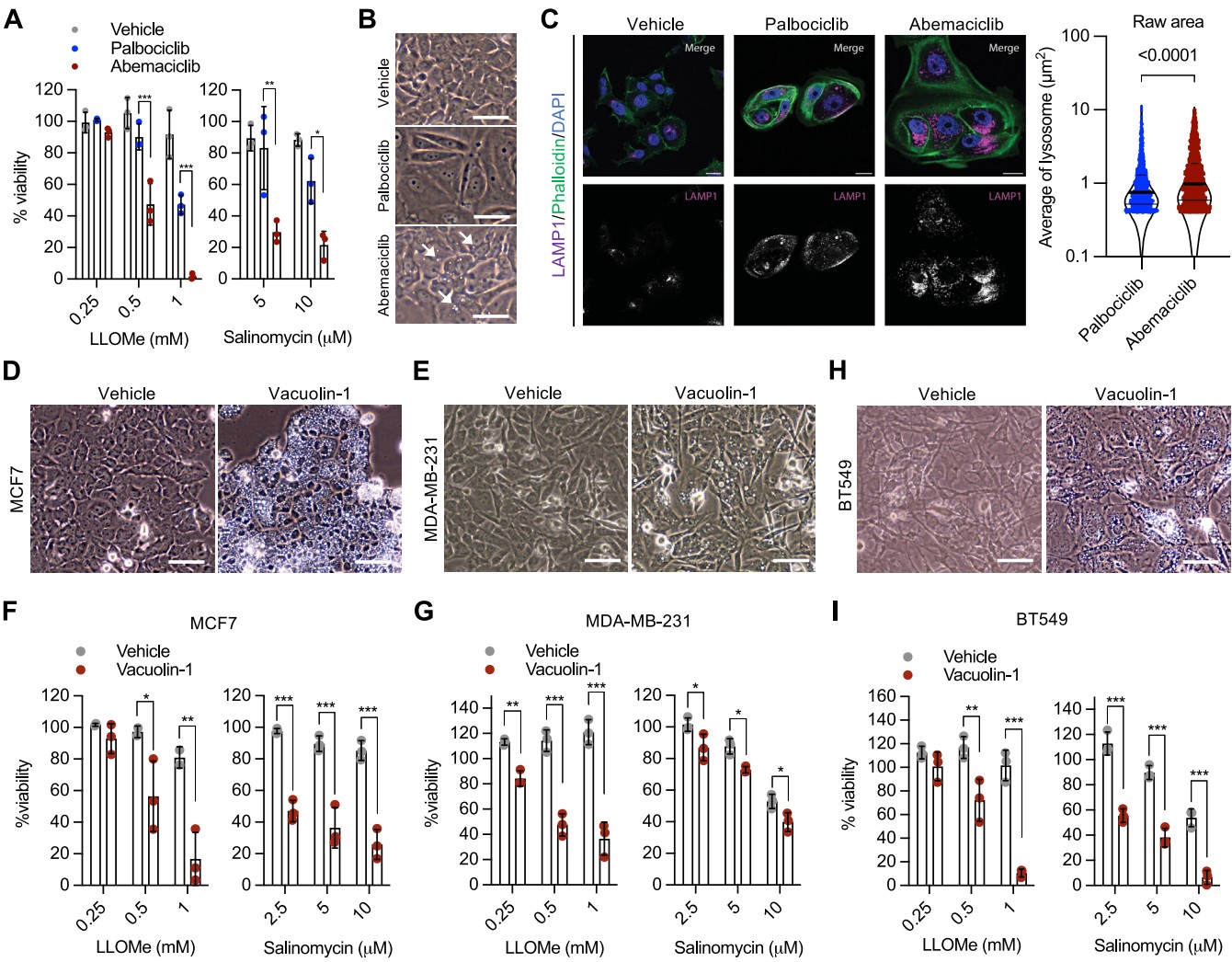

**Figure 5. Lysosomal enlargement enhances the sensitivity to lysosomotropic agent-induced cell death.**

(A–C) MCF-7 cells were treated with vehicle (water), palbociclib (1 μM for 8 days), or abemaciclib (1 μM for 8 days), then replated and subsequently treated with LLOMe ($p < 0.0001$ for all) or salinomycin ($p_{5μM} = 0.0025$, $p_{10μM} = 0.0191$). Cell viability was assessed using the MTS assay (A). Cells were also imaged to identify vacuoles (white arrow; scale bar, 200 μm) (B), stained with LAMP1 (scale bar, 10 μm), and lysosomal size was quantified ($p < 0.0001$) (C). $n = 3$ independent experiments. (D–G) Cells were treated with vehicle or vacuolin-1, then imaged to visualize vacuole formation (D) for MCF-7, and (E) for MDA-MB-231 (scale bar, 200 μm), or subsequently treated with LLOMe or salinomycin, and cell viability was quantified using the MTS assay (F) for MCF-7 (LLOMe $p_{0.5mM} = 0.0341$, $p_{1mM} = 0.0023$; salinomycin all concentration $p < 0.0001$), (G) for MDA-MB-231 (LLOMe $p_{0.25mM} = 0.0049$, other concentrations $p < 0.0001$; salinomycin $p_{2.5μM} = 0.0179$, $p_{5μM} = 0.0159$, $p_{10μM} = 0.0339$). $n = 3$ independent experiments. (H, I) BT549 cells were treated with vehicle or vacuolin-1, then imaged for vacuoles visualization (scale bar, 200 μm) (H), or subsequently treated with LLOMe or salinomycin and cell viability was evaluated using MTS assay (LLOMe $p_{0.5mM} = 0.0011$, $p_{1mM} < 0.001$; salinomycin all concentration $p < 0.0001$) (I), $n = 3$ independent experiments. Data are shown as the mean ± SD, two-way ANOVA. *$p < 0.05$, **$p < 0.01$, ***$p < 0.001$. Source data are available online for this figure.

mitotic status. Importantly, analysis of breast cancer biopsies collected from human patients before and after treatment showed a CDK4/6i-induced increase in lysosomal mass. Additionally, we observed an enlargement in the lysosomal size. Lysosomal enlargement has been suggested to increase the sensitivity to lysosomotropic agent-induced cell death through lysosomal membrane permeabilization (Wang et al, 2018). We showed that a sequential therapeutic approach involving abemaciclib followed by lysosomotropic agents such as LLOMe and salinomycin was effective in inducing cell death in different HR⁺ breast cancer cells. Due to the fact that RB1 integrity is a common predictor of

abemaciclib sensitivity, we investigated its role in lysosomal changes. We demonstrated that RB1 knockdown in abemaciclib-treated breast cancer cells resulted in reduced lysosomal biogenesis, accompanied by decreased sensitivity to lysosomotropic agent-induced cell death triggered by abemaciclib. These observed effects on lysosomal biogenesis may be directly or indirectly linked to RB activity. Suppression of RB can potentially inhibit lysosomal biogenesis at the RNA level through other downstream factors. For instance, CDK1 suppression of TFEB activity suggests that active cell cycling can inhibit TFEB (Odle et al, 2020). However, other mechanisms that underlie this phenomenon cannot be

excluded. In addition to HR[+] breast cancer cells, abemaciclib treatment significantly increased lysosomal content and sensitivity to lysosomotropic agents in MDA-MB-231 cells, a TNBC cell line that is responsive to CDK4/6 inhibition. Conversely, BT549 cells demonstrated resistance to both abemaciclib and the subsequent lysosomotropic agent treatment. These findings highlight the heterogeneity within TNBC and underscore the need for precision medicine approaches that consider the specific cellular and molecular characteristics of different tumor types.

Previous studies have shown that abemaciclib promotes aberrant lysosomal vacuolization and general lysosomal enlargement (Hino et al, 2020). Interestingly, our data indicated that vacuolization was sufficient and necessary to sensitize breast cancer cells to the toxic effects of lysosomotropic agents, suggesting the potential for the development of more precise sequential therapeutic approaches that combine drugs promoting vacuolization with lysosomotropic agents. Therefore, identification of markers that can predict treatment efficacy is critical. It is logical to assume that an increase in lysosomal biogenesis and, potentially, lysosomal vacuolization may predict an increased sensitivity to cell death by compounds that can cause LMP. However, lysosomotropic agents induce LMP through different mechanisms of action, which rely on additional changes that follow the first treatment. Thus, further studies are needed to evaluate the involvement of different mechanisms that sensitize cancer cells to a specific type of treatment, allowing for a more refined prediction of treatment efficacy.

Overall, the sequential combinatorial strategy presented here supports the enormous potential of targeting aberrant lysosomal biology for the treatment of breast cancer. This strategy is distinct from traditional approaches targeting DNA or broad cellular pathways and offers an innovative solution to overcome the limitations of CDK4/6 inhibitor monotherapy and to improve treatment outcomes.

# Methods

### Reagents and tools table

| Reagent/Resource | Reference or Source | Identifier or Catalog Number |
|---|---|---|
| **Experimental models** | | |
| MCF-7 (*H. sapiens*) | ATCC | HTB-22 |
| BT474 (*H. sapiens*) | ATCC | HTB-20 |
| T47D (*H. sapiens*) | ATCC | HTB-133 |
| ZR-75-30 (*H. sapiens*) | ATCC | CRL-1504 |
| MDA-MB-231 (*H. sapiens*) | ATCC | CRM-HTB-26 |
| BT549 (*H. sapiens*) | ATCC | HTB-122 |
| BJ (*H. sapiens*) | ATCC | CRL-2522 |
| IMR90 (*H. sapiens*) | ATCC | CCL-186 |
| HEK293FT (*H. sapiens*) | ATCC | CRL-3216 |
| Phoenix-AMPHO (*H. sapiens*) | ATCC | CRL-3213 |
| Female *Foxn1*[Nu] mice (*M. musculus*) | Charles River | Crl:NU(NCr)-*Foxn1*[nu] (490) |

| Reagent/Resource | Reference or Source | Identifier or Catalog Number |
|---|---|---|
| Tumor samples from patients (*H. sapiens*) | This study | N/A |
| **Recombinant DNA** | | |
| pLenti6-H2b-mCherry | Addgene | #89766 |
| FU-H2B-GFP-IRES-Puro | Addgene | #69550 |
| pLenti-mCherry-CDK4KTR | Addgene | #126680 |
| pMRX-IP TFEB-sfGFP | Addgene | #135402 |
| ViraPower plasmid mix | Invitrogen | K497-00 |
| **Antibodies** | | |
| LAMP1 | Bioke | 9091S |
| CD63 | Novus biologicals | NBP2-42225 |
| Anti-rabbit-AlexaFluor633 | Thermo Fisher | A-21072 |
| Phalloidin-AlexaFluor488 | AAT Bioquest | 23115 |
| **Oligonucleotides and other sequence-based reagents** | | |
| qPCR primers | This study | Table 1 |
| siRB1 Silencer Select | Thermo Fisher | 4390824 |
| siNTC Silencer Select | Thermo Fisher | 4390843 |
| **Chemicals, Enzymes and other reagents** | | |
| Abemaciclib | MedChemExpress | HY-16297 |
| Palbociclib | MedChemExpress | HY-50767 |
| L-Leucyl-L-Leucine methyl ester | Cayman Chemical | 16008 |
| Salinomycin | MedChemExpress | HY-15597 |
| Bafilomycin-A1 | MedChemExpress | HY-100558 |
| Q-VD-OPH | MedChemExpress | HY-12305 |
| Navitoclax (ABT-263) | MedChemExpress | HY-10087 |
| Dasatinib hydrochloride | MedChemExpress | HY-10181A |
| Quercetin hydrate | Sigma-Aldrich | 337951 |
| Fisetin | MedChemExpress | HY-N0182 |
| Nutlin-3a | MedChemExpress | HY-10029 |
| Piperlongumine | MedChemExpress | HY-N2329 |
| Geldanamycin (HSP90 inhibitor) | Cayman Chemical | 13355 |
| Vacuolin-1 | MedChemExpress | HY-118630 |
| CellTox™ Green Cytotoxicity Assay | Promega | G8741 |
| CellTiter 96® Aqueous Non-Radiative Cell Proliferation Assay | Promega | G5421 |
| Acridine orange | Sigma-Aldrich | 235474 |
| CytoFix™ Red Lysosomal Stain | AAT Bioquest | 23210 |
| TurboFect™ Transfection Reagent | Thermo Fisher | R0532 |
| DAPI | Sigma-Aldrich | D5942 |
| Isolate II RNA Mini Kit | Bioline | BIO-52073 |
| Applied Biosystems™ High-Capacity cDNA Reverse Transcription Kit | Applied Biosystems | 4368813 |
| GoTaq® qPCR Master Mix | Promega | A6002 |
| 17β-estradiol pellets | Innovative Research of America | E-121 |

| Reagent/Resource | Reference or Source | Identifier or Catalog Number |
|---|---|---|
| **Software** | | |
| ImageJ | NIH | N/A |
| Prism 10 | GraphPad | N/A |
| Powerpoint | Microsoft | N/A |
| R | R-Project | Version 4.2.3 |
| **Other** | | |
| IncuCyte® ZOOM system | Essen BioScience | N/A |

## Cell culture and treatments

MCF-7, BT474, T47D, ZR-75-30, MDA-MB-231, BT549, IMR90, and BJ cells were purchased from ATCC. All cells were cultured in DMEM-glutaMAX pyruvate medium (Thermo Fisher) supplemented with 10% fetal bovine serum (Thermo Fisher) and 1% penicillin-streptomycin (Lonza). Cells were kept in 5% $O_2$, 5% $CO_2$, 90% $N_2$, and 37 °C incubators and were regularly tested for mycoplasma contamination.

For treatment, abemaciclib (MedChemExpress, HY-16297) was dissolved in water and diluted in DMEM media to a final concentration of 1 µM. Palbociclib (MedChemExpress, HY-50767), vacuolin-1 (MedChemExpress, HY-118630), lysosomotropic agents L-leucyl-l-leucine methyl ester (Cayman, 16008), and salinomycin (MedChemExpress, HY-15597) were dissolved in DMSO. Bafilomycin-A1 (MedChemExpress, HY-100558) and Q-VD-OPH (MedChemExpress, HY-12305) were dissolved in DMSO. All drugs were further diluted in DMEM to treat the cells at different concentrations, as indicated in each figure. Senolytic drugs were dissolved in DMSO and further diluted in DMEM to treat cells at different concentrations, as indicated in Fig. 1: navitoclax (ABT-263) (MedChemExpress, HY-10087), dasatinib hydrochloride (MedChemExpress, HY-10181A), quercetin hydrate (Sigma-Aldrich, 337951), fisetin (MedChemExpress, HY-N0182), Nutlin-3a (MedChemExpress, HY-10029), piperlongumine (MedChemExpress, HY-N2329), and geldanamycin (HSP90 inhibitor) (Cayman Chemical, 13355).

## Colony formation assay

Drug-treated breast cancer cells were plated in a 6-well plate (5 × $10^3$ cells/well) at the end of treatment and allowed to grow in drug-free normal medium for eight days. Next, the cells were fixed in 4% PFA for 30 min and stained with 0.2% crystal violet in 37% methanol for 1 h. Images of all plates were taken using a scanner (Epson). The images were cropped and processed using Microsoft PowerPoint using the same settings.

## EdU staining

Control and treated cells were replated on coverslips in a 24-well plate (3 × $10^4$ cells/well) and cultured for 20 h in the presence of EdU (10 µM), and then fixed and stained as previously described (Kohli et al, 2021). Images were acquired at 100× magnification

(Leica), and the number of cells was counted using the ImageJ software.

## Senescence-associated β-galactosidase staining

Control and treated cells were replated in a 24-well plate (2 × $10^4$ cells/well). After 24 h, the cells were fixed and incubated with Sa-β-galactosidase stain as previously described (Kohli et al, 2021). Images were acquired using a Leica microscope and processed using ImageJ software.

## MTS viability assay

Cells were seeded in 96-well plates. Drug treatments were performed directly on the plate and three technical replicates were used for each experimental condition. After 24 or 48 h of treatment, cell viability was measured using MTS solution (CellTiter 96® Aqueous Non-Radiative Cell Proliferation Assay, Promega) mixed with the culture medium, following the manufacturer's instructions. The absorbance was recorded at 450 nm and normalized to the reference absorbance measured in the acellular wells.

## Lentivirus production and transduction

To produce lentiviral particles, HEK293FT cells were plated in a 10 cm Petri dish. 24 h later, the cells were transfected with a mix of plasmids encoding viral components (ViraPower plasmid mix, K497-00, Invitrogen) in addition to the lentiviral plasmid of interest using TurboFect™ Transfection Reagent (R0532, Thermo Fisher) overnight. The next day, the transfection mix was replaced with normal growth media, collected 24 or 48 h later, and used immediately for transducing cells or frozen at −80 °C. The pLenti6-H2b-mCherry plasmid was a gift from Torsten Wittmann (Addgene, plasmid #89766), FU-H2B-GFP-IRES-Puro plasmid was a gift from Charles Gersbach (Addgene, plasmid #69550), pLenti-mCherry-CDK4KTR plasmid was a gift from Hee Won Yang (Addgene, plasmid #126680). Lentiviral particles harboring H2b-mCherry or H2B-GFP constructs were used to generate cells with stable nuclear mCherry or GFP, respectively. Lentiviral particles harboring the mCherry-CDK4KTR construct were used to generate cells expressing a stable CDK4/6 kinase activity reporter. For transduction, cells were plated in 6-well plate and once attached, viral particles were added in addition to normal growth media, and the plate was centrifuged at 4000 × $g$ for 40 min. The next day, the cells were washed and refreshed with complete medium, and two days later, the cells were selected with blasticidin (2.5 µM).

## Retrovirus production and transduction

To produce retroviral particles, Phoenix-AMPHO cells were plated in a 10 cm Petri dish. 24 h later, cells were transfected with pMRX-IP TFEB-sfGFP plasmid, a gift from Eisuke Itakura (Addgene, plasmid #135402), using the TurboFect™ Transfection Reagent (R0532, Thermo Fisher) overnight. The next day, the transfection mix was replaced with normal growth medium, collected 48 h later, and used immediately to transduce the recipient cells. The TFEB-GFP-labeled cells were then selected using puromycin.

## Real-time imaging of cell proliferation and death

Time-lapse imaging was performed using the IncuCyte® ZOOM system to examine cell death and proliferation in real-time. Cells stably expressing a nuclear-localized red fluorescent (H2b-mcherry-labeled), were cultured in the presence of Celltox Green, a non-toxic cell-impermeable dye that emits green fluorescence once it binds to nucleic acids. Therefore, once the plasma membrane is permeabilized, the dye can enter the cell and bind to nuclear DNA. Double-positive cells (green and red) were considered dead. The number of red and double-positive cells was automatically quantified using the IncuCyte ZOOM software.

## Acridine orange staining

For Acridine Orange staining, cells were first washed and then incubated with 1 μM acridine orange (Sigma-Aldrich) in PBS for ~20 min at 37 °C. The cells were then rinsed and stored in a complete medium or PBS during image acquisition. Images were acquired in red and green channels using a Leica microscope. Images were processed using ImageJ software.

## LAMP1 cellular staining

Immunofluorescence for LAMP1 was conducted by seeding cells on a glass coverslip (Aurion 72231-01) at 60% cell density after senescence induction. Cells were fixed with 4% PFA and washed three times with PBS, followed by a blocking step with 0.2-micron filtered 3% BSA and 0.15% glycine in PBS supplemented with 0.1% saponin for 30 min at room temperature. Next, the samples were stained with LAMP1 Ab (Bioke, 9091S) overnight at 4 °C, followed by three washes with PBS and consecutive secondary staining with anti-rabbit-AlexaFluor633 (Thermo Fisher Scientific, A-21072), phalloidin-AlexaFluor488 (AAT Bioquest, 23115), and DAPI (Sigma-Aldrich, D5942-5MG). After three PBS washes to remove unbound reagents, the samples were mounted on glass microscopy slides and imaged using SP8(. ×) at 63× magnification with an N/A of 1.4. Image analysis was conducted with an ImageJ script (see Computer Code EV1), with further processing of the data using pandas in Python.

## Flow cytometry

Cells were incubated with 1 μg/ml Acridine Orange (Sigma-Aldrich, 235474-5G) or a 1:1000 dilution of pre-dissolved CytoFix Red Lysosomal Stain (AAT Bioquest, 23210) in DMEM with 10% FBS for 30 min at 37 C and 5% $CO_2$. Next, the cells were trypsinized from the plate and washed twice in cold PBS with 1% bovine serum albumin (BSA) before measuring the samples on Canto II. Flow cytometry analysis was performed using the FlowJo V10. For LAMP1 staining, cells were fixed after being trypsinized from the plate and fixed with 4% PFA, followed by a protocol described in the section 'Lysosomal analysis by microscopy,' but only using the anti-rabbit-AlexaFluor633 as secondary.

## Bioinformatics

We performed the gene expression analysis using the 'DESeq2' package (Love et al, 2014), applying an alpha cutoff of 0.05 and using the Bonferroni $p$-value adjustment method. We performed Gene Set Enrichment Analysis (GSEA) based on the fold-change values obtained from DESeq2 and visualized the results using the 'gseGO' function from the 'clusterProfiler' package (Wu et al, 2021). A volcano plot was generated using the 'EnhancedVolcano' package. All RNA-seq data were analyzed using R version 4.2.3.

## Real-time PCR

The Isolate II RNA Mini Kit (Bioline, Cat# BIO-52073) was used for total RNA isolation. For reverse transcription, 500 ng RNA was transcribed into cDNA using a kit (Applied Biosystems, Cat# 4368813). qRT-PCR was performed using GoTaq® qPCR Master Mix (A6002, Promega). Tubulin was used as a housekeeping gene to normalize the expression of target genes. The primer sequences are listed in Table 1.

## Spheroids

Breast cancer spheroids were constructed as described previously (Soto-Gamez et al, 2022). Briefly, cells were seeded at a density of 1500 cells/well in ultra-low attachment 96-well plates (Corning Incorporated, Kennebunk, ME, USA). The plates were centrifuged at 1000 rpm for 5 min to promote cell aggregation and spheroid formation. After 6 days of incubation, spheroids were suitable for performing experiments, and cell senescence was induced by treatment with abemaciclib (1 μM) for 6 additional days. The spheroids were then treated with salinomycin (5 μM) or vehicle (DMSO) for 48 h. Cell death was monitored in a time-resolved assay using CellTox™ Green Cytotoxicity Assay (Promega) to measure fluorescence increases every 2 h with an IncuCyte S3 (Essen BioScience). The mean green fluorescence intensity relative to the initial time point (0h00m), and area under the curve (AUC) for each of the conditions tested were calculated.

## Mouse xenografts

All the mice were maintained in individually ventilated cages (IVC) at the Central Animal Facility (CDP) of the University Medical Center Groningen (UMCG) under standard conditions. The experiments were approved by the Central Authority for Scientific Procedures on Animals (CCD) of the Netherlands. 12-week-old female *Foxn1*[Nu] (Charles River) mice were used to generate MCF-7 and MDA-MB-231 xenograft cancer models.

17β-estradiol pellets (0.18 mg for 90-day release, catalog number E-121, Innovative Research of America) were transplanted into the left flank of 12-week-old female *Foxn1*[Nu] mice, and MCF-7 cells ($2 \times 10^6$) were injected into the right flank. Approximately 3 weeks after injection, the mice were randomly divided into four groups and injected with PBS or abemaciclib (40 mg/kg) for 7 consecutive days, followed by vehicle or salinomycin (5 mg/kg) for an additional 7 consecutive days. Tumor growth was calculated by measuring the tumor length and width using a caliper, and the investigator was blinded. Mice exhibiting skin side effects due to estrogen pellets were excluded from the experiment before the start of treatment injections.

MDA-231 cells ($2 \times 10^6$) were injected into the right mammary fat pads of 12-week-old female *Foxn1*[Nu] mice. Approximately 3 weeks after injection, the mice were divided into four groups and injected with PBS or abemaciclib (40 mg/kg) for 7 consecutive days,

**Table 1.** List of qPCR primers.

| Genes | Species | Direction | Sequence |
|---|---|---|---|
| TUBULIN | Human | Forward | CTTCGTCTCCGCCATCAG |
| | | Reverse | CGTGTTCCAGGCAGTAGAGC |
| HPRT1 | Human | Forward | TGACCTTGATTTATTTTGCATACC |
| | | Reverse | CGAGCAAGACGTTCAGTCCT |
| CDKN1A | Human | Forward | TCACTGTCTTGTACCCTTGTGC |
| | | Reverse | GGCGTTTGGAGTGGTAGAAA |
| RB1 | Human | Forward | CTTCCTCATGCTGTTCAGGAG |
| | | Reverse | TGCATGAAGACCGAGTTATAGAAT |
| IL6 | Human | Forward | CAGGAGCCCAGCTATGAACT |
| | | Reverse | GAAGGCAGCAGGCAACAC |
| CXCL1 | Human | Forward | CATCGAAAAGATGCTGAACAGT |
| | | Reverse | ATAAGGGCAGGGCCTCCT |
| CCL2 | Human | Forward | AGTCTCTGCCGCCCTTCT |
| | | Reverse | AGTCTCTGCCGCCCTTCT |
| IGFBP3 | Human | Forward | AACGCTAGTGCCGTCAGC |
| | | Reverse | CGGTCTTCCTCCGACTCAC |
| LIF | Human | Forward | TGAAGTGCAGCCCATAATGA |
| | | Reverse | TTCCAGTGCAGAACCAACAG |
| LAMP1 | Human | Forward | CGTGTCACGAAGGCGTTTTCAG |
| | | Reverse | CTGTTCTCGTCCAGCAGACACT |
| LAMP2 | Human | Forward | TTTAACTAAAAACAAAAGTTCCCAAAG |
| | | Reverse | CTTGGAATGAATAACCAACTCACTT |
| GAA | Human | Forward | TCTACAGCGTGGAGTTCTCCGA |
| | | Reverse | GCTGAAGGAACTGGTCCGCAAA |
| CD63 | Human | Forward | CCGGCAGCAGATGGAGAATT |
| | | Reverse | GTGTAGTTAGCAGCCCCACA |
| CTSD | Human | Forward | GCAAACTGCTGGACATCGCTTG |
| | | Reverse | GCCATAGTGGATGTCAAACGAGG |
| HEXA | Human | Forward | GGAGGTCATTGAATACGCACGG |
| | | Reverse | GGATTCACTGGTCCAAAGGTGC |
| MCOLN1 | Human | Forward | CGGACTGCTATACCTTCAGCGT |
| | | Reverse | GGTGCTTACACTCCTGGATGTGT |

hormone therapy with aromatase inhibitors. All samples were collected according to a protocol approved by the Italian Local Ethics Committee (CEROM IRST IRCCS-AVR; protocol code: IRST B114), and all patients provided written informed consent. The procedure was performed before starting the treatment and at the time of the best clinical response, respectively, 6.9, 11.8, and 7.6 months after starting treatment. At the last follow-up, three patients were disease-free. Samples were stained with LAMP1 primary antibody (9091S, Bioke) and CD63 primary antibody (NBP2-42225, Novus Biologicals) overnight at 4 °C, followed by three washes with PBS and subsequent staining with secondary antibodies and DAPI (Sigma-Aldrich, D5942-5MG). After the final three PBS washes to remove unbound reagents, the samples were mounted and imaged at 100× magnification (Leica).

## Ethical approval

Mouse experiments were approved by the Central Authority for Scientific Procedures on Animals (CCD) of the Netherlands. All human samples were collected according to a protocol approved by the Italian Local Ethics Committee (CEROM IRST IRCCS-AVR; protocol code: IRST B114), and all patients provided written informed consent.

## Statistical analysis

GraphPad Prism 10 was used for statistical analyses. Detailed information is provided in the figure legends.

# Data availability

No data amenable to large-scale repository deposition were generated in this study. The data are provided in the manuscript or supplementary information files. All datasets and access numbers used in this study are cited in their relative descriptions. Additional reasonable requests can be made to the corresponding authors directly.

The source data of this paper are collected in the following database record: biostudies:S-SCDT-10_1038-S44318-025-00371-x.

# Peer review information

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

followed by vehicle or salinomycin (5 mg/kg) for an additional 7 consecutive days. Tumor growth was calculated by measuring the tumor length and width using a caliper, and the investigator was blinded. After excision, the weight of each tumor was measured. The weight of each mouse was measured every week.

## Patients

Formalin fixed paraffin embedded tumor samples from three hormone receptor-positive, HER2 negative metastatic breast cancer patients were collected at IRCCS Istituto Romagnolo per lo Studio dei Tumori (IRST) "Dino Amadori," Meldola, Italy before and after exposure to treatment with CDK4/6 inhibitors (one treated with abemaciclib, one with ribociclib and one with palbociclib) and

the lipid kinase PIKfyve proceeds through lysosome coalescence. J Cell Sci 131:jcs213587

Curnock R, Yalci K, Palmfeldt J, Jaattela M, Liu B, Carroll B (2023) TFEB-dependent lysosome biogenesis is required for senescence. EMBO J 42:e111241

Demaria M, O'Leary MN, Chang J, Shao L, Liu S, Alimirah F, Koenig K, Le C, Mitin N, Deal AM et al (2017) Cellular senescence promotes adverse effects of chemotherapy and cancer relapse. Cancer Discov 7:165–176

Eggersmann TK, Degenhardt T, Gluz O, Wuerstlein R, Harbeck N (2019) CDK4/6 inhibitors expand the therapeutic options in breast cancer: palbociclib, ribociclib and abemaciclib. BioDrugs 33:125–135

Evangelou K, Belogiannis K, Papaspyropoulos A, Petty R, Gorgoulis VG (2023) Escape from senescence: molecular basis and therapeutic ramifications. J Pathol 260:649–665

Ewald JA, Desotelle JA, Wilding G, Jarrard DF (2010) Therapy-induced senescence in cancer. J Natl Cancer Inst 102:1536–1546

Fassl A, Brain C, Abu-Remaileh M, Stukan I, Butter D, Stepien P, Feit AS, Bergholz J, Michowski W, Otto T et al (2020) Increased lysosomal biomass is responsible for the resistance of triple-negative breast cancers to CDK4/6 inhibition. Sci Adv 6:eabb2210

Goel S, DeCristo MJ, Watt AC, BrinJones H, Sceneay J, Li BB, Khan N, Ubellacker JM, Xie S, Metzger-Filho O et al (2017) CDK4/6 inhibition triggers anti-tumour immunity. Nature 548:471–475

Hafner M, Mills CE, Subramanian K, Chen C, Chung M, Boswell SA, Everley RA, Liu C, Walmsley CS, Juric D et al (2019) Multiomics profiling establishes the polypharmacology of FDA-approved CDK4/6 inhibitors and the potential for differential clinical activity. Cell Chem Biol 26:1067–1080.e1068

Hernandez-Segura A, Nehme J, Demaria M (2018) Hallmarks of cellular senescence. Trends Cell Biol 28:436–453

Hino H, Iriyama N, Kokuba H, Kazama H, Moriya S, Takano N, Hiramoto M, Aizawa S, Miyazawa K (2020) Abemaciclib induces atypical cell death in cancer cells characterized by formation of cytoplasmic vacuoles derived from lysosomes. Cancer Sci 111:2132–2145

Hu M, Carraway 3rd KL (2020) Repurposing cationic amphiphilic drugs and derivatives to engage lysosomal cell death in cancer treatment. Front Oncol 10:605361

Hu Y, Gao J, Wang M, Li M (2021) Potential prospect of CDK4/6 inhibitors in triple-negative breast cancer. Cancer Manag Res 13:5223–5237

Kallunki T, Olsen OD, Jaattela M (2013) Cancer-associated lysosomal changes: friends or foes? Oncogene 32:1995–2004

Kavcic N, Butinar M, Sobotic B, Hafner Cesen M, Petelin A, Bojic L, Zavasnik Bergant T, Bratovs A, Reinheckel T, Turk B (2020) Intracellular cathepsin C levels determine sensitivity of cells to leucyl-leucine methyl ester-triggered apoptosis. FEBS J 287:5148–5166

Kohli J, Wang B, Brandenburg SM, Basisty N, Evangelou K, Varela-Eirin M, Campisi J, Schilling B, Gorgoulis V, Demaria M (2021) Algorithmic assessment of cellular senescence in experimental and clinical specimens. Nat Protoc 16:2471–2498

Kulbay M, Paimboeuf A, Ozdemir D, Bernier J (2022) Review of cancer cell resistance mechanisms to apoptosis and actual targeted therapies. J Cell Biochem 123:1736–1761

Lee S, Schmitt CA (2019) The dynamic nature of senescence in cancer. Nat Cell Biol 21:94–101

Li W, Kawaguchi K, Tanaka S, He C, Maeshima Y, Suzuki E, Toi M (2023) Cellular senescence triggers intracellular acidification and lysosomal pH alkalinized via ATP6AP2 attenuation in breast cancer cells. Commun Biol 6:1147

Llanos S, Megias D, Blanco-Aparicio C, Hernandez-Encinas E, Rovira M, Pietrocola F, Serrano M (2019) Lysosomal trapping of palbociclib and its functional implications. Oncogene 38:3886–3902

Love MI, Huber W, Anders S (2014) Moderated estimation of fold change and dispersion for RNA-seq data with DESeq2. Genome Biol 15:550

Mai TT, Hamai A, Hienzsch A, Caneque T, Muller S, Wicinski J, Cabaud O, Leroy C, David A, Acevedo V et al (2017) Salinomycin kills cancer stem cells by sequestering iron in lysosomes. Nat Chem 9:1025–1033

Maltoni R, Roncadori A, Balzi W, Mazza M, Nicolini F, Palleschi M, Ulivi P, Bravaccini S (2024) An Italian real-world study highlights the importance of some clinicopathological characteristics useful in identifying metastatic breast cancer patients resistant to CDK4/6 inhibitors and hormone therapy. Biomedicines 12:498

Martinez-Carreres L, Puyal J, Leal-Esteban LC, Orpinell M, Castillo-Armengol J, Giralt A, Dergai O, Moret C, Barquissau V, Nasrallah A et al (2019) CDK4 regulates lysosomal function and mTORC1 activation to promote cancer cell survival. Cancer Res 79:5245–5259

McGrath MK, Abolhassani A, Guy L, Elshazly AM, Barrett JT, Mivechi NF, Gewirtz DA, Schoenlein PV (2024) Autophagy and senescence facilitate the development of antiestrogen resistance in ER positive breast cancer. Front Endocrinol 15:1298423

O'Brien N, Conklin D, Beckmann R, Luo T, Chau K, Thomas J, Mc Nulty A, Marchal C, Kalous O, von Euw E et al (2018) Preclinical activity of abemaciclib alone or in combination with antimitotic and targeted therapies in breast cancer. Mol Cancer Ther 17:897–907

Odle RI, Walker SA, Oxley D, Kidger AM, Balmanno K, Gilley R, Okkenhaug H, Florey O, Ktistakis NT, Cook SJ (2020) An mTORC1-to-CDK1 switch maintains autophagy suppression during mitosis. Mol Cell 77:228–240.e227

O'Leary B, Finn RS, Turner NC (2016) Treating cancer with selective CDK4/6 inhibitors. Nat Rev Clin Oncol 13:417–430

Rocca A, Farolfi A, Bravaccini S, Schirone A, Amadori D (2014) Palbociclib (PD 0332991): targeting the cell cycle machinery in breast cancer. Expert Opin Pharmacother 15:407–420

Rocca A, Schirone A, Maltoni R, Bravaccini S, Cecconetto L, Farolfi A, Bronte G, Andreis D (2017) Progress with palbociclib in breast cancer: latest evidence and clinical considerations. Ther Adv Med Oncol 9:83–105

Rovira M, Sereda R, Pladevall-Morera D, Ramponi V, Marin I, Maus M, Madrigal-Matute J, Diaz A, Garcia F, Munoz J et al (2022) The lysosomal proteome of senescent cells contributes to the senescence secretome. Aging Cell 21:e13707

Sano O, Kazetani K, Funata M, Fukuda Y, Matsui J, Iwata H (2016) Vacuolin-1 inhibits autophagy by impairing lysosomal maturation via PIKfyve inhibition. FEBS Lett 590:1576–1585

Schmitt CA, Wang B, Demaria M (2022) Senescence and cancer—role and therapeutic opportunities. Nat Rev Clin Oncol 19:619–636

Sieben CJ, Sturmlechner I, van de Sluis B, van Deursen JM (2018) Two-step senescence-focused cancer therapies. Trends Cell Biol 28:723–737

Soto-Gamez A, Wang Y, Zhou X, Seras L, Quax W, Demaria M (2022) Enhanced extrinsic apoptosis of therapy-induced senescent cancer cells using a death receptor 5 (DR5) selective agonist. Cancer Lett 525:67–75

Thiele DL, Lipsky PE (1990) Mechanism of L-leucyl-L-leucine methyl ester-mediated killing of cytotoxic lymphocytes: dependence on a lysosomal thiol protease, dipeptidyl peptidase I, that is enriched in these cells. Proc Natl Acad Sci USA 87:83–87

Wang B, Demaria M (2021) The quest to define and target cellular senescence in cancer. Cancer Res 81:6087–6089

Wang B, Varela-Eirin M, Brandenburg SM, Hernandez-Segura A, van Vliet T, Jongbloed EM, Wilting SM, Ohtani N, Jager A, Demaria M (2022) Pharmacological CDK4/6 inhibition reveals a p53-dependent senescent state with restricted toxicity. EMBO J 41:e108946

Wang F, Gomez-Sintes R, Boya P (2018) Lysosomal membrane permeabilization and cell death. Traffic 19:918–931

Wang L, Bernards R (2018) Taking advantage of drug resistance, a new approach in the war on cancer. Front Med 12:490–495

Watt AC, Cejas P, DeCristo MJ, Metzger-Filho O, Lam EYN, Qiu X, BrinJones H, Kesten N, Coulson R, Font-Tello A et al (2021) CDK4/6 inhibition reprograms the breast cancer enhancer landscape by stimulating AP-1 transcriptional activity. Nat Cancer 2:34–48

Wilkinson L, Gathani T (2022) Understanding breast cancer as a global health concern. Br J Radio 95:20211033

Wu T, Hu E, Xu S, Chen M, Guo P, Dai Z, Feng T, Zhou L, Tang W, Zhan L et al (2021) clusterProfiler 4.0: a universal enrichment tool for interpreting omics data. Innovation 2:100141

Yao Z, Murali B, Ren Q, Luo X, Faget DV, Cole T, Ricci B, Thotala D, Monahan J, van Deursen JM et al (2020) Therapy-induced senescence drives bone loss. Cancer Res 80:1171–1182

Yin Q, Jian Y, Xu M, Huang X, Wang N, Liu Z, Li Q, Li J, Zhou H, Xu L et al (2020) CDK4/6 regulate lysosome biogenesis through TFEB/TFE3. J Cell Biol 219:e201911036

## Acknowledgements

The project was funded by grants to the laboratory of M.D. from the Nederlandse Organisatie voor Wetenschappelijk Onderzoek (NWO, VIDI #09150172010029), the Dutch Cancer Foundation (KWF, #14547) and the Hevolution Foundation (HF-GRO-23-1199094-17).

## Author contributions

**Jamil Nehme**: Conceptualization; Investigation; Methodology; Writing—original draft; Writing—review and editing. **Sjors Maassen**: Data curation; Methodology. **Sara Bravaccini**: Resources; Investigation. **Michele Zanoni**: Resources; Data curation. **Caterina Gianni**: Resources. **Ugo De Giorgi**: Resources. **Abel Soto-Gamez**: Investigation; Methodology. **Abdullah Altulea**: Resources; Data curation; Formal analysis. **Teodora Gheorghe**: Data curation; Methodology. **Boshi Wang**: Resources; Supervision; Investigation; Methodology; Writing—original draft; Writing—review and editing. **Marco Demaria**: Conceptualization; Resources; Supervision; Funding acquisition; Writing—original draft; Project administration; Writing—review and editing.

Source data underlying figure panels in this paper may have individual authorship assigned. Where available, figure panel/source data authorship is listed in the following database record: biostudies:S-SCDT-10_1038-S44318-025-00371-x.

## Disclosure and competing interests statement

M.D. is founder and shareholder of Cleara Biotech and advisor for Oisin Biotechnologies and Rubedo Life Sciences. The M.D. laboratory received funding from Ono Pharmaceuticals. None of the companies mentioned above were involved in this study. The remaining authors declare no competing interests.

# Expanded View Figures

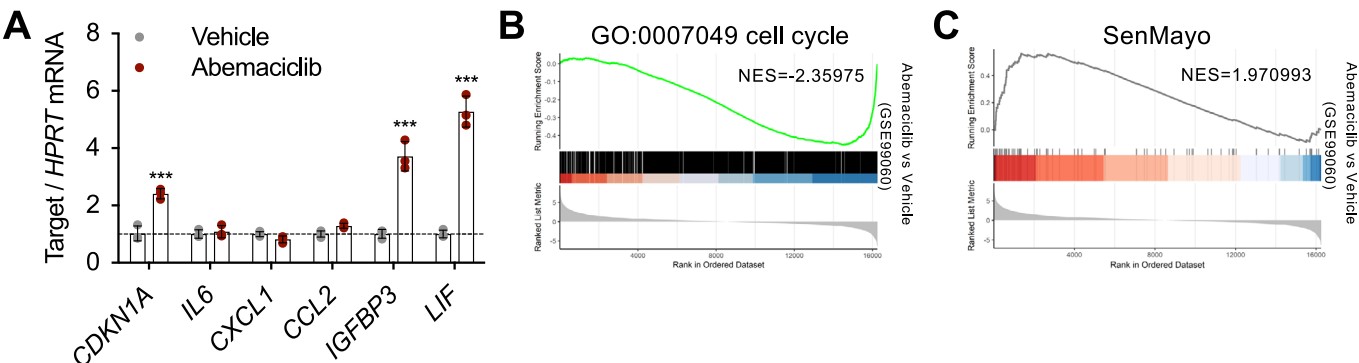

**Figure EV1.  Abemaciclib treatment induces senescence-like phenotypes in MCF-7 breast cancer cells.**

(**A**) RNA was extracted from MCF-7 cells treated with either the vehicle (water) or abemaciclib (1 µM for 8 days), followed by qPCR analysis targeting the specified genes. $n = 3$ independent experiments, $p < 0.0001$. (**B**, **C**) Gene Set Enrichment Analysis (GSEA) plot showing the enrichment for the GO term "cell cycle" (**B**) and the SenMayo geneset (**C**) in MCF-7 cells treated with abemaciclib compared to vehicle-treated cells. The expression data were obtained from GSE99060. Data are mean ± SD. For (**A**): two-way ANOVA; ***$p < 0.001$.

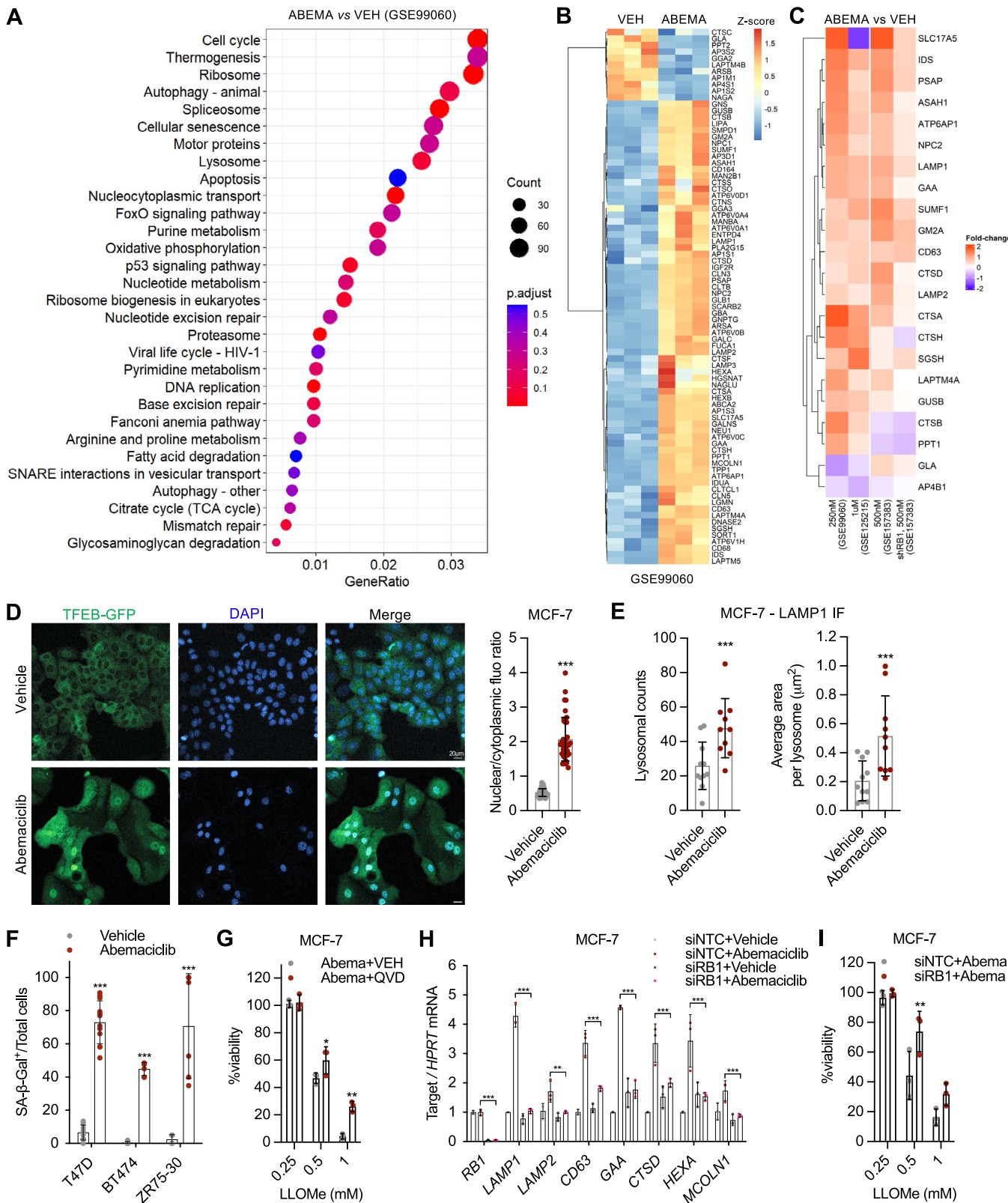

**Figure EV2.   Abemaciclib treatment increases lysosomal mass in HR$^+$ breast cancer cells.**

(A) Dot plot shows the significantly enriched GO terms when comparing the gene expression profiles of MCF-7 cells treated with abemaciclib to those of vehicle-treated cells. Hypergeometric test (clusterProfiler package) for dot plots. (B) Heatmap of differentially expressed lysosomal genes following abemaciclib treatment of MCF-7 cells. The expression data were obtained from GSE99060. (C) Heatmap of differentially expressed lysosomal genes following abemaciclib treatment (fold change *vs.* vehicle) in MCF-7 cells with the indicated doses and genetic backgrounds. Expression data were obtained from the GSE99060, GSE125215, and GSE157383 datasets. (D) MCF-7 cells labeled with TFEB-GFP via lentiviral particles were treated with either vehicle (water) or abemaciclib (1 μM for 48 h), followed by imaging to assess the subcellular localization of TFEB proteins (scale bar: 20 μm). The ratio of nuclear-to-cytoplasmic fluorescence signals was quantified and plotted. $n = 3$ independent experiments; $p < 0.0001$. (E) MCF-7 cells were treated with either vehicle (water) or abemaciclib (1 μM for 8 days), replated, and stained for LAMP1. The cells were then analyzed to quantify the lysosomal counts and average lysosomal area. The cells were obtained from three independent experiments ($p < 0.0001$). (F) Quantification of SA-β-Gal positive cells from Fig. 2J, $p < 0.0001$ for all. (G) MCF-7 cells were treated with abemaciclib, with or without QVD (5 μM), followed by treatment with LLOMe at the indicated concentrations. Cell viability was measured using the MTS assay. $n = 3$ independent experiments, $p_{0.5mM} = 0.0311$, $p_{1mM} = 0.009$. (H) MCF-7 cells were transfected with siRNA targeting *RB1* and then treated with vehicle or abemaciclib (1 μM for 5 days) 1 d after siRNA transfection. At the end of the treatment, RNA was extracted from both siNTC and siRB1 groups, and qPCR was performed to assess the expression of *RB1* and lysosomal genes. $n = 3$ independent experiments, $p_{LAMP2} = 0.005$, and other genes $p < 0.001$. (I) MCF-7 cells were transfected with siRNA targeting *RB1* and treated with either vehicle or abemaciclib (1 μM for 5 days), starting 1 day after siRNA transfection. Following this, the cells were replated and subjected to subsequent treatment with LLOMe for 48 h at the indicated concentrations. $n = 3$ independent experiments, $p_{0.5mM} = 0.0028$. Data are mean ± SD. For (D, E), unpaired Student's *t*-tests (two-tailed) were used. For (F–I), two-way ANOVA was used. *$p < 0.05$, **$p < 0.01$, ***$p < 0.001$.

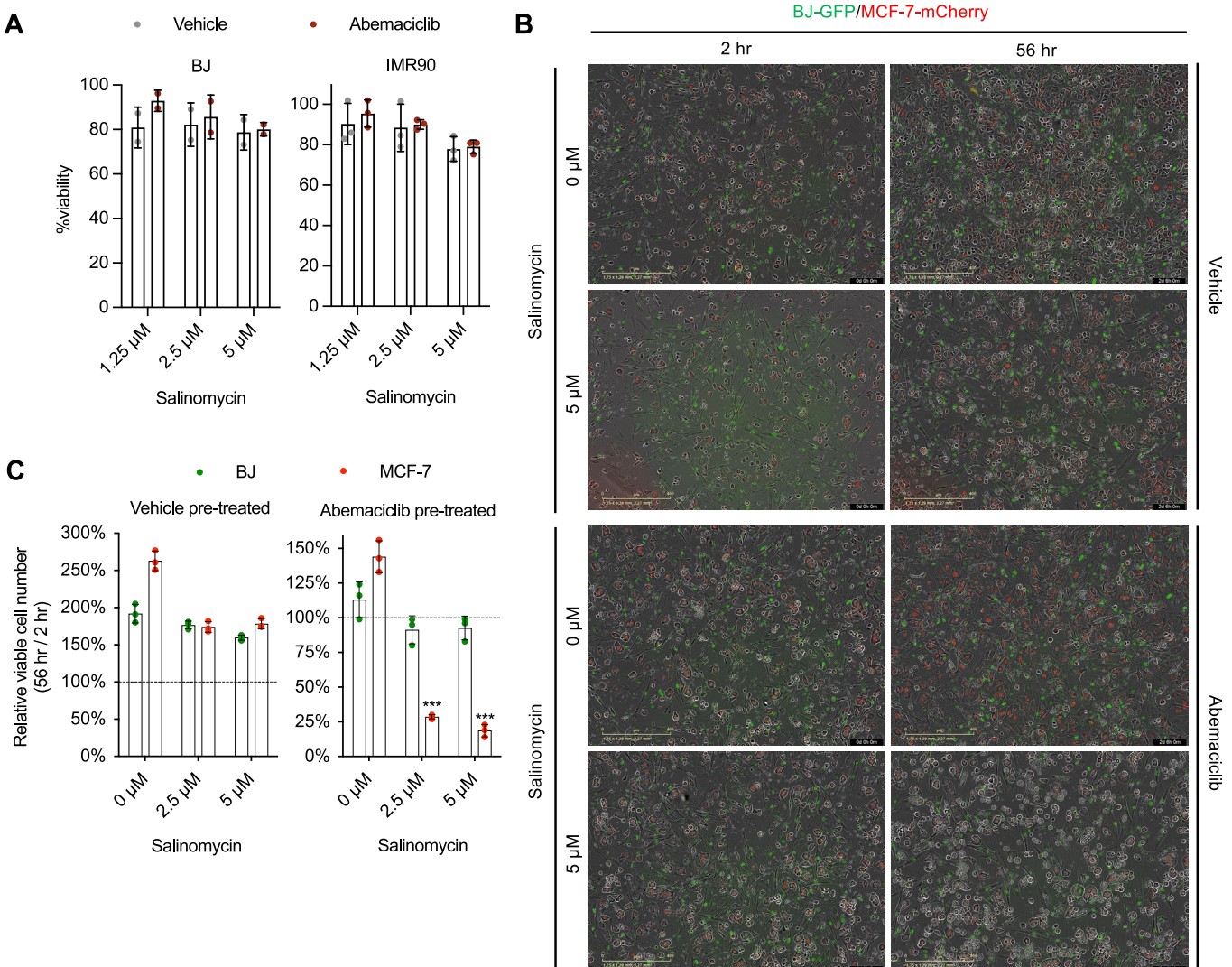

**Figure EV3. Abemaciclib-treated normal fibroblasts are not sensitive to lysosomotropic agents.**

(A) BJ and IMR90 cells were pretreated with either vehicle (water) or abemaciclib (1 μM for 7 days), followed by incubation with salinomycin at the indicated concentrations. Cell viability was assessed using an MTS assay. $n = 3$ independent experiments. (B) MCF-7 cells (mCherry) and BJ cells (GFP) were pretreated with either vehicle (water) or abemaciclib (1 μM for six days) and then co-cultured. These cells were subsequently incubated with salinomycin (0 or 5 μM for 56 h) and representative images were captured using IncuCyte at 2 and 56 h. (C) Ratio of viable MCF-7 and BJ cells (56-hr *vs* 2-hr) treated with different concentrations of salinomycin for both vehicle- and abemaciclib-pretreated groups. $n = 3$ independent experiments. $p < 0.001$ for all experiments. Data are mean ± SD. Two-way ANOVA. ***$p < 0.001$.

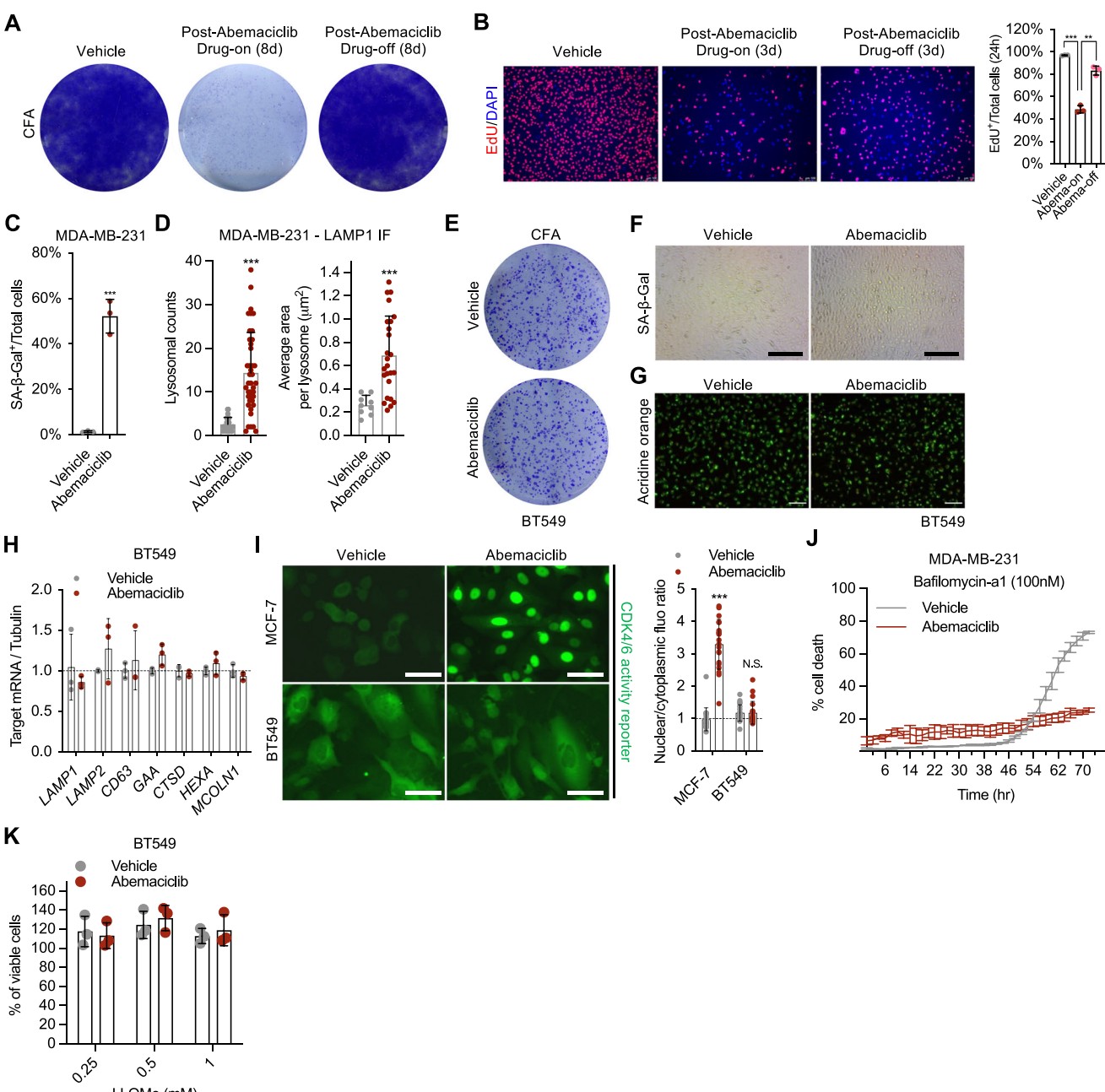

◀

**Figure EV4. Abemaciclib selectively sensitizes triple-negative breast cancer cells to lysosomotropic agents-induced cell death.**

(A, B) MDA-MB-231 cells were treated with either vehicle (water) or abemaciclib (1 μM for 8 days) and then replated for subsequent assays. For the colony formation assay (A), cells were cultured with or without abemaciclib for an additional eight days, followed by staining. For EdU staining (B), cells were replated and treated with or without abemaciclib for 3 days. EdU (10 μM) was added on day 4 and incubated for 20 h, followed by staining (scale bar, 100 μm). $n = 3$ independent experiments, $p_{on-vs-veh} < 0.0001$; $p_{on-vs-off} = 0.0027$. (C) Vehicle- or abemaciclib (1 μM for 8 days)-treated MDA-MB-231 cells were stained with LAMP1 and analyzed for lysosomal count and average area. $n = 3$ independent experiments; $p < 0.0001$. (D) MDA-MB-231 cells were treated with vehicle (water) or abemaciclib (1 μM for 8 days), replated and stained with LAMP1, and the cells were quantified for lysosomal counts or average area of lysosomes. The cells were obtained from three independent experiments ($p < 0.0001$). (E–H) BT549 cells were treated with vehicle (water) or abemaciclib (1 μM for 8 days), and then cells were replated for colony formation assay (8 days culture) (E), SA-β-Gal staining (scale bar, 1 mm) (F), acridine orange staining (scale bar, 1 mm) (G), or qRT-PCR for lysosomal genes (H), $n = 3$ independent experiments. (I) MCF-7 and BT549 cells were transfected with lentiviral particles encoding a CDK4/6 kinase activity reporter (CDK4KTR), followed by treatment with either vehicle or abemaciclib (1 μM for 48 h). Cells were imaged to analyze the localization of the mCherry protein (scale bar: 60 μm). Cytoplasmic localization of mCherry indicates active CDK4/6 kinases, whereas nuclear localization indicates suppressed CDK4/6 kinase activity. The nuclear-to-cytoplasmic mCherry fluorescence mean intensity ratio was calculated and plotted for each cell. The data represent cells from three independent experiments ($p < 0.001$). (J) Vehicle-or abemaciclib (1 μM for 8 days)-pretreated MDA-MB-231 cells were subsequently treated with bafilomycin A1, and cell death was measured using IncuCyte live cell imaging with Celltox™ Green. $n = 3$ independent experiments. (K) Vehicle- or abemaciclib-pretreated BT549 cells were subsequently treated with LLOMe and viability was measured using the MTS assay. $n = 3$ independent experiments. Data are mean ± SD. For (B), one-way ANOVA. For (C, D), unpaired Student's $t$-test (two-tailed) was used. For (H–K), two-way ANOVA was used. ***$p < 0.001$.

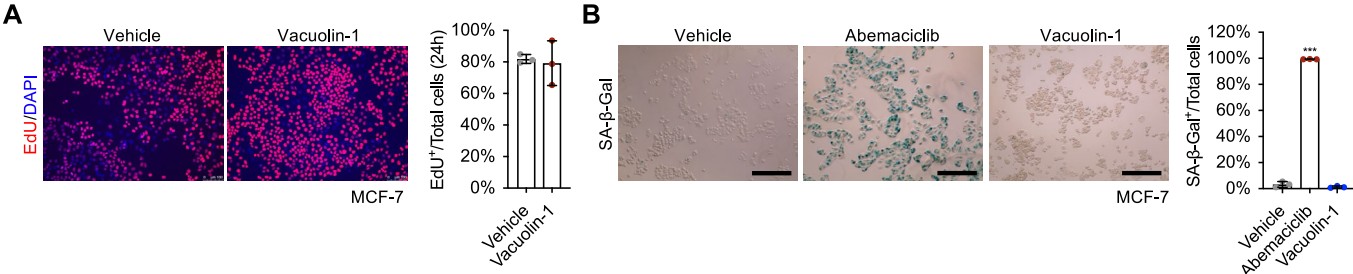

**Figure EV5.  Vacuolin-1 is unable to induce senescence-like phenotype.**

(A, B) MCF-7 cells were treated with vehicle or vacuolin-1 (1 μM for 8 days) and re-plated for EdU staining (scale bar, 100 μm) (A) or SA-β-Gal staining (scale bar, 1 mm) (B). *n* = 3 independent experiments; *p* < 0.0001. Data are presented as the mean ± standard deviation (SD), one-way ANOVA. \*\*\**p* < 0.001.

