## [Peer Review File · The EMBO Journal]

Pharmacological CDK4/6 inhibition promotes vulnerability to lysosomotropic agents in breast cancer

Jamil Nehme, Sjors Maassen, Sara Bravaccini, Michele Zanoni, Caterina Gianni, Ugo De Giorgi, Abel Soto-Gamez, Abdullah Altulea, Teodora Gheorghe, Boshi Wang, and Marco Demaria

Corresponding authors: Marco Demaria (m.demaria@umcg.nl) , Boshi Wang (b.wang01@umcg.nl)

Review Timeline:

Submission Date:	4th Jul 24
Editorial Decision:	17th Aug 24
Revision Received:	2nd Dec 24
Editorial Decision:	16th Dec 24
Revision Received:	9th Jan 25
Accepted:	10th Jan 25

Editor: Daniel Klimmeck

Transaction Report:

Dear Dr Marco Demaria,

Thank you again for the submission of your manuscript (EMBOJ-2024-118394) to The EMBO Journal. Please accept my apologies for getting back to you with unusual delay due to protracted referee input and detailed discussion in the editorial team. As mentioned earlier, your study was assessed by three reviewers with expertise in cancer biology and translational senescence, whose comments are enclosed below.

As you will see from the experts' reports, the referees acknowledge the analysis and potential interest of your results. However, they also express major concerns regarding depth and completeness of the findings, which need to be addressed thoroughly to make them supportive of publication in the EMBO Journal. The reviewers also raise a number of issues related to the data presentation, additional controls and improved methods annotation required, statistics applied and overall discussion of related literature, that would need to be conclusively addressed to achieve the level of robustness and clarity needed for The EMBO Journal.

Given the overall interest stated and broader angle of your findings, we are able to invite you to revise your manuscript experimentally to address the referees' comments. I need to stress though that we do require strong support from the referees on a revised version of the study in order to move on to publication of the work.

In light of the extensive experimentation requested, I would appreciate if you could contact me during the next weeks for exchange e.g. a video call to discuss your perspective on the comments and potential plan for revisions.

Please feel free to contact me if you have any questions or need further input on the referee comments.

When submitting your revised manuscript, please carefully review the instructions below.

Please feel free to approach me any time should you have additional questions related to this.

Thank you for the opportunity to consider your work for publication.

I look forward to your revision.

Best regards,

Daniel Klimmeck

Daniel Klimmeck, PhD
Senior Editor
The EMBO Journal

Instruction for the preparation of your revised manuscript:

- 1) a .docx formatted version of the manuscript text (including legends for main figures, EV figures and tables). Please make sure that the changes are highlighted to be clearly visible.
- 2) individual production quality figure files as .eps, .tif, .jpg (one file per figure).
- 3) a .docx formatted letter INCLUDING the reviewers' reports and your detailed point-by-point response to their comments. As part of the EMBO Press transparent editorial process, the point-by-point response is part of the Review Process File (RPF), which will be published alongside your paper.
- 4) a complete author checklist, which you can download from our author guidelines (<https://wol-prod-cdn.literatumonline.com/pb->

assets/embo-site/Author Checklist%20-%20EMBO%20J-1561436015657.xlsx). Please insert information in the checklist that is also reflected in the manuscript. The completed author checklist will also be part of the RPF.

6) It is mandatory to include a 'Data Availability' section after the Materials and Methods. Before submitting your revision, primary datasets produced in this study need to be deposited in an appropriate public database, and the accession numbers and database listed under 'Data Availability'. Please remember to provide a reviewer password if the datasets are not yet public (see <https://www.embopress.org/page/journal/14602075/authorguide#datadeposition>).

7) Our journal encourages inclusion of *data citations in the reference list* to directly cite datasets that were re-used and obtained from public databases. Data citations in the article text are distinct from normal bibliographical citations and should directly link to the database records from which the data can be accessed. In the main text, data citations are formatted as follows: "Data ref: Smith et al, 2001" or "Data ref: NCBI Sequence Read Archive PRJNA342805, 2017". In the Reference list, data citations must be labeled with "[DATASET]". A data reference must provide the database name, accession number/identifiers and a resolvable link to the landing page from which the data can be accessed at the end of the reference. Further instructions are available at .

8) At EMBO Press we ask authors to provide source data for the main and EV figures. Our source data coordinator will contact you to discuss which figure panels we would need source data for and will also provide you with helpful tips on how to upload and organize the files.

Numerical data can be provided as individual .xls or .csv files (including a tab describing the data). For 'blots' or microscopy, uncropped images should be submitted (using a zip archive or a single pdf per main figure if multiple images need to be supplied for one panel). Additional information on source data and instruction on how to label the files are available at .

9) We replaced Supplementary Information with Expanded View (EV) Figures and Tables that are collapsible/expandable online (see examples in <https://www.embopress.org/doi/10.15252/emj.201695874>). A maximum of 5 EV Figures can be typeset. EV Figures should be cited as 'Figure EV1, Figure EV2' etc. in the text and their respective legends should be included in the main text after the legends of regular figures.

11) For data quantification: please specify the name of the statistical test used to generate error bars and P values, the number (n) of independent experiments (specify technical or biological replicates) underlying each data point and the test used to calculate p-values in each figure legend. The figure legends should contain a basic description of n, P and the test applied. Graphs must include a description of the bars and the error bars (s.d., s.e.m.).

The revision must be submitted online within 90 days; please click on the link below to submit the revision online before 15th Nov 2024.

Referee #1:

Nehme and colleagues report a series of very well-controlled experiments that unequivocally demonstrate that CDK4/6 inhibitor-treated breast cancer cells develop a vulnerability to lysosomotropic agents, such as L-leucyl-L-leucine methyl ester (LLOMe) and salinomycin. They propose a new drug combination to treat both HR+ and TNBC. This is a high-quality and potentially translatable study. The main weakness is the lack of a direct mechanism linking CDK4/6 inhibition to the increased expression of lysosomal genes. The RB mutant cell line BT549 did not respond to the treatment combination suggesting that RB mediates the effect likely by repressing E2F target genes. However, it is important to note that this cell line has many other alterations that could explain the lack of response. Inactivating RB in any of the sensitive cell lines would provide considerable insights into the mechanism behind the acquired vulnerability to lysosomotropic agents in breast cancer cells.

Minor points:

- 1- Quantification of % of SAbGal + cells in figure 1a and 2j.
- 2- Is cell death induced by lysosomotropic agents in this study caspase-dependent?
- 3- Serrano and colleagues reported localization of CDK4i into lysosomes (PMID: 30692638). Did the authors verify if a similar localization of CDK4i occurred in their system? It could be interesting to know if lysosomal trapping was a factor explaining resistance in BT549 cells.
- 4- Does vacuolin-1 also sensitize normal cells to CDK4i?

Referee #2:

In this manuscript Nehme and co-workers identify lysosomal changes in various hormone receptor + (HR+) breast cancer (BC) cell lines following treatment with the Cdk4/6 inhibitor abemaciclib, resulting in increased expression of lysosome-related genes accompanied by increased lysosome numbers and volume. Importantly, they observed increased expression of the lysosomal marker LAMP1 in paired biopsies from three BC patients following treatment with aromatase and various Cdk4/6 inhibitors. The authors went on to demonstrate that these lysosomal alterations represent a therapeutic vulnerability by combining abemaciclib with lysosomotropic agents in 2D and 3D (spheroids) assays. The most efficacious combination (abemaciclib + salinomycin) efficiently reduced the growth of subcutaneous implants of MCF7 cells when compared to single agent treatments. This combination was also validated (both in vitro and in vivo) in a triple negative BC cell line; the observed therapeutic benefit appears to be RB-dependent. Finally, the authors demonstrate that the sensitivity to lysosomotropic agents in vitro is not restricted to Cdk4/6 inhibitors but also applicable to other conditions resulting in enhanced lysosome vacuolization such as Vacuolin-1.

The manuscript is clearly written and the overall technical quality of the data presented in the manuscript is satisfactory and sufficiently supports their claims. Nevertheless, the reported findings are not entirely novel. Abemaciclib has been shown to enhance the expression of genes related to lysosomal biogenesis inducing lysosome dysfunction in a BC mouse model (doi:10.1158/0008-5472.CAN-19-0708). Similarly, the enhanced efficacy of Cdk4/6 inhibitors in BC by co-treatment with lysosomotropic agents has also been reported (doi:10.1126/sciadv.abb2210). These manuscripts should be cited. Having said this, the utilization of in vivo experiments together with a limited number of paired human biopsies in the current manuscript is interesting.

I have a number of questions that, in the opinion of this reviewer, should be addressed or discussed before the manuscript is ready for publication.

1. The authors exclusively show increased expression of the lysosome marker LAMP1 in the post Cdk4/6 biopsies as a surrogate marker for increased lysosomal mass. While this is correct, it would be interesting to further document this phenotype by measuring additional lysosome markers (such as those shown in Fig 2C) by RT-PCR if the authors have access to additional FFPE material.
2. Following on patient data, in addition to my previous request, or as an alternative approach, the authors could use publicly available RNAseq data from BC patients treated with aromatase + Cdk4/6 inhibitors to investigate whether enhanced expression

of lysosomal genes and lysosomal-associated pathways is a general feature. As an example please see doi:10.1038/s41523-024-00625-7 for data availability.

3. There is limited information regarding the BC biopsies used in the study. At least the authors should mention whether the post-Cdk4/6 biopsy was collected at the time of clinical response or at progression.

4. While the phenotypic consequences of combining Cdk4/6i and lysosomotropic agents are well characterized in vitro, this is not the case in vivo (Figures 3J/K & 4J/K) where the efficacy is exclusively assessed by measuring tumour size and mass. At least the authors should evaluate whether in vivo the response is cytostatic or cytotoxic by performing immunostaining/immunofluorescence of apoptotic markers in tumour sections following the different treatments.

Referee #3:

The manuscript by Nehme and collaborators studies the connection between CDK4/6 inhibitors and lysosomes in breast cancer cells, investigating the role of cellular senescence in this context. The authors describe that treatment with Abemaciclib causes changes in lysosome mass and structure in breast cancer cells, and show that these lysosomal alterations lead to increased sensitivity to lysosomotropic agents. The manuscript addresses an interesting question with potential translational implications. However, in its present form, the advancement relative to the current knowledge seems limited. The general link of cellular senescence to the lysosomal compartment is well established. Also, Abemaciclib (and other CDK4/6i) have previously been connected with lysosome function and senescence in several tumor cell lines, including some used in this study (Martinez-Carreres 2019 PMID 31395606, Li 2023 PMID 37993606, Rovira 2022 PMID 36087066), and the sensitivity of senescent cells (Curnock 2023 PMID 36970883) or CDK4/6i-treated cells (Llanos 2019 PMID 30692638) to lysosomotropic agents has been reported. A potentially relevant contribution of the study would be to establish whether or not this effect of CDK4/6i on lysosomes is necessarily associated to cellular senescence. Unfortunately, the current data about this question is somewhat preliminary and more work is needed to support the conclusions.

Major points

1. The potential link to senescence is an important issue of this study. Previous studies have reported that Abema, and other CDK4/6i, can induce a full senescent phenotype and senolytic sensitivity in tumor cell lines (Rovira 2022, Martinez-Carreres 2019, Li 2023 and others). In this manuscript, the authors conclude that the effect of CDK4/6i on lysosomes can be dissociated from full senescence, based on the apparent lack of cell-cycle arrest in MDA-MB-231 cells. This is a critical point that should be characterized in more detail. Late time points in the cell count experiment (Figure ED3a) seem to indicate reduced proliferation with Abema and the images of the clonogenic assays (Figure ED3b) are not very informative. More sensitive assays, such as EdU incorporation, should be used to assess proliferation, and markers of senescence arrest, like p21, should also be monitored.

As pointed out by the authors, it is possible that the treatment regimens used might account for the contrasting results on proliferation/senescence from different studies. It would be interesting if the authors could address this question directly in their experimental settings. For instance, what is the impact of Abema on proliferation in MDA-MB-231 cells maintained in drug-containing medium compared to the current conditions where drug is withdrawn before the functional assays?

In this context, it would be important to characterize better the senescence-like phenotype induced by Abema in MCF7 cells (Figure 1). Additional senescence markers should be monitored by QPCR, IF or WB. It would also be interesting to know if they display a specific SASP profile as previously described by the authors in Abema-treated non-transformed fibroblasts (Wang 2022 PMID 34985783).

If the difference in cell-cycle arrest is confirmed, it would be interesting to know if this is a TNBC-specific phenotype. Do the other HR+ cell lines show a full senescent phenotype, similarly to MCF7?

2. The authors conclude that Abema-treated BJ normal fibroblasts are not sensitive to Salinomycin based on the data from a co-culture experiment with MCF7 cells (Figure ED2). How do they interpret this result considering their own report (Wang 2022) that Abema renders BJ fibroblasts senescent with gene expression enrichment in lysosomal-associated pathways? The potential difference between cancer and normal cells is an important question that deserves more attention. It would be more informative to perform assays in BJ cells alone, analogous to the ones shown for breast cancer cells (expression of lysosomal markers, sensitivity to senolytics and lysosomotropic drugs).

3. Figure 5 shows that vacuolin increases the lysosomal compartment and induces vulnerability to lysosomotropic drugs. Vacuolization has been associated with senescence. Did vacuolin treatment have any effect on senescence markers?

4. The manuscript would be improved if the authors could elaborate more on the potential mechanism. The results with the Rb-negative BT549 cell line (Fig ED 3) suggest that the impact of CDK4/6i on lysosomes is linked to the on-target effect on the cell-

cycle machinery. In addition, the data in Figure 5 suggest a critical role of vacuolization. It would be interesting to discuss whether these two processes might be connected. Interestingly, other reports have suggested that the effect of CDK4/6i on lysosomes involves mTOR signaling (Martinez-Carreres 2019). This potential mechanism should also be discussed and, ideally, tested experimentally with pharmacological perturbation of this pathway.

5. Several relevant articles currently not mentioned in the manuscript should be cited and discussed: Llanos 2019 PMID 30692638, Martinez-Carreres 2019 PMID 31395606, Rovira 2022 PMID 36087066, Li 2023 PMID 37993606, Curnock 2023 PMID 36970883 among others.

Minor points

1. The current display and description in the text of differential gene expression data from an independent study (Goel 2017 PMID 28813415) is somewhat misleading (Figures 1c, d, 2a, b and ED1a, b). It should be clearly stated in the main text that this is not original data from the current study. There is also a concern that the conditions used in both studies do not appear to be the same (1 μ M Abemaciclib for 8 days in the current manuscript versus 250 nM for 7 days in Goel 2017).

2. In the lysosome stainings in Figure 2d and ED 1c, d, please show quantifications and similar magnification examples for all the markers.

3. In the spheroid assay (Figure 3j), Salinomycin alone causes significant cell death but this effect is not obvious in the tumorigenesis assay (Figure 3k). How do the authors interpret these results? Also, Abema alone seems to have a negligible effect in the tumorigenesis assays (Figure 3k). Wouldn't one expect to see a reduction in tumor growth consistent with senescence induction?

4. I recommend using the standard name MDA-MB-231 (Cellosaurus CVCL_0062) to designate the cell line called MDA-231 in the manuscript.

5. Please clarify if the indicated number of samples are biological or technical replicates.

Referee #1:

Nehme and colleagues report a series of very well-controlled experiments that unequivocally demonstrate that CDK4/6 inhibitor-treated breast cancer cells develop a vulnerability to lysosomotropic agents, such as L-leucyl-L-leucine methyl ester (LLOMe) and salinomycin. They propose a new drug combination to treat both HR+ and TNBC. This is a high-quality and potentially translatable study. The main weakness is the lack of a direct mechanism linking CDK4/6 inhibition to the increased expression of lysosomal genes. The RB mutant cell line BT549 did not respond to the treatment combination suggesting that RB mediates the effect likely by repressing E2F target genes. However, it is important to note that this cell line has many other alterations that could explain the lack of response. Inactivating RB in any of the sensitive cell lines would provide considerable insights into the mechanism behind the acquired vulnerability to lysosomotropic agents in breast cancer cells.

AUTHORS: We would like to thank the reviewer for the insightful feedback.

To further clarify the mechanism involved, we conducted additional experiments as follows:

1. **TFEB Localization:** TFEB, along with TFE3, are key transcription factors that regulate lysosomal gene biogenesis. Previous studies have shown that CDK4/6 phosphorylates TFEB, leading to its exclusion from the nucleus. Therefore, inhibiting CDK4/6 is expected to keep TFEB within the nucleus, enhancing its transcriptional activity. To test this, we used MCF7 cells expressing TFEB tagged with GFP. Our observations confirmed an increase in TFEB's nuclear localization when CDK4/6 was inhibited, suggesting that the increase in lysosomal mass is likely due to the activation of TFEB-dependent transcription. Please refer to figure EV 2D.
2. **Effect of RB1 Knockdown on Lysosomal Biogenesis:** We knocked down RB1 and assessed its impact on lysosomal gene transcription. We observed a decrease in lysosomal biogenesis, which correlated with reduced cell sensitivity to death induced by lysosomotropic agents after treatment with abemaciclib. This suggests that mechanisms downstream of RB suppression may inhibit lysosomal biogenesis at the RNA level. One potential explanation is the role of CDK1, which has been implicated in the suppression of TFEB activity, suggesting that continued cell cycling may inhibit TFEB and thereby reduce lysosomal biogenesis. Please refer to figure EV 2 H and I.
3. **Resistance to CDK4/6 Inhibition in BT549 cells:** Additionally, to investigate why BT549 cells did not develop the lysosomal phenotype, we examined whether this cell line responds to CDK4/6 inhibition. Using a CDK4/6 activity probe, we found that BT549 cells do not exhibit any response to CDK4/6 inhibition, and CDK4/6 activity remains unaffected. This suggests the presence of upstream mechanisms of resistance. Potential explanations for this resistance could include multidrug resistance mechanisms, restricted accessibility to CDK4/6, or other yet unidentified pathways. Please refer to figure EV 4.

Minor points:

1- Quantification of % of SAbGal + cells in figure 1a and 2j.

The percentage of SA- β -gal positive cells was quantified, and the data is presented in figure 1A, EV 2F, and EV4C for reference.

2- Is cell death induced by lysosomotropic agents in this study caspase-dependent?

To assess the dependency of cell death on caspase activity, we utilized a pan-caspase inhibitor. The results revealed only partial reliance on caspase activity for cell death, indicating the involvement of additional mechanisms beyond caspase-dependent pathways. Please refer to figure EV 2G

3- Serrano and colleagues reported localization of CDK4i into lysosomes (PMID: 30692638). Did the authors verify if a similar localization of CDK4i occurred in their system? It could be interesting to know if lysosomal trapping was a factor explaining resistance in BT549 cells.

This has not been evaluated, as the primary focus of this study was not on the resistance of cells to abemaciclib. However, it is an interesting hypothesis that could be explored in future experiments.

4- Does vacuolin-1 also sensitize normal cells to CDK4i?

Normal cells are intrinsically sensitive to CDK4/6 inhibitors in the context of cell cycle regulation. Furthermore, the potential role of vacuolin as a sensitizer to enhance the effects of CDK4/6 inhibition has not been investigated in this study.

Referee #2:

In this manuscript Nehme and co-workers identify lysosomal changes in various hormone receptor + (HR+) breast cancer (BC) cell lines following treatment with the Cdk4/6 inhibitor abemaciclib, resulting in increased expression of lysosome-related genes accompanied by increased lysosome numbers and volume. Importantly, they observed increased expression of the lysosomal marker LAMP1 in paired biopsies from three BC patients following treatment with aromatase and various Cdk4/6 inhibitors. The authors went on to demonstrate that these lysosomal alterations represent a therapeutic vulnerability by combining abemaciclib with lysosomotropic agents in 2D and 3D (spheroids) assays. The most efficacious combination (abemaciclib + salinomycin) efficiently reduced the growth of subcutaneous implants of MCF7 cells when compared to single agent treatments. This combination was also validated (both in vitro and in vivo) in a triple negative BC cell line; the observed therapeutic benefit appears to be RB-dependent. Finally, the authors demonstrate that the sensitivity to lysosomotropic agents in vitro is not restricted to Cdk4/6 inhibitors but also applicable to other conditions resulting in enhanced lysosome vacuolization such as Vacuolin-1.

The manuscript is clearly written and the overall technical quality of the data presented in the manuscript is satisfactory and sufficiently supports their claims. Nevertheless, the reported findings are not entirely novel. Abemaciclib has been shown to enhance the expression of genes related to lysosomal biogenesis inducing lysosome dysfunction in a BC mouse model (doi:10.1158/0008-5472.CAN-19-0708). Similarly, the enhanced efficacy of Cdk4/6 inhibitors in BC by co-treatment with lysosomotropic agents has also been reported (doi:10.1126/sciadv.abb2210). These manuscripts should be cited. Having said this, the utilization of in vivo experiments together with a limited number of paired human biopsies in the current manuscript is interesting.

I have a number of questions that, in the opinion of this reviewer, should be addressed or discussed before the manuscript is ready for publication.

We would like to thank the reviewer for their valuable comments. We hope our responses and discussions have adequately addressed their concerns.

Indeed, these two papers reported an increase in lysosomal mass following CDK4/6 inhibition. However, our study primarily focused on inducing cell death in cancer cells using lysosomotropic agents, rather

than simply observing changes in lysosomal mass. Unlike our study, these papers did not explore or highlight cell death using lysosomotropic agents. Instead, they used lysosomotropic agents to deacidify lysosomes and prevent resistance to CDK4/6 inhibitors.

It is also important to note that lysosomotropic agents are not all the same and can produce different cellular responses despite all targeting lysosomes. For instance, in our study, we found that bafilomycin, which reduces lysosomal acidity, did not affect cell viability. Furthermore, while these studies used a co-treatment approach where the lysosomotropic agents were applied from the start, our approach was sequential, adding the agents only after the phenotype had begun to develop.

In the study titled "CDK4 regulates lysosomal function and mTORC1 activation to promote cancer cell survival" (Martínez-Carreres et al 2019), the researchers used a combination therapy from the beginning of the treatment. Specifically, cells were cultured for 8 days with a combination of an AMPK activator and a CDK4 inhibitor, leading to approximately 30% of cells becoming Annexin V-positive. This approach is fundamentally different from ours and, therefore, cannot be considered a senolytic therapy. Moreover, the cell death observed in their study was autophagic and linked to impaired lysosomal function, which is in contrast with our findings. In our study, we demonstrated that bafilomycin A1, which deacidifies lysosomes and reduces their functionality, was unable to kill cells pre-exposed to abemaciclib. This further highlights the distinct mechanisms at play in our research compared to the approach taken in their study.

We have now cited these papers in the discussion section.

1. The authors exclusively show increased expression of the lysosome marker LAMP1 in the post Cdk4/6 biopsies as a surrogate marker for increased lysosomal mass. While this is correct, it would be interesting to further document this phenotype by measuring additional lysosome markers (such as those shown in Fig 2C) by RT-PCR if the authors have access to additional FFPE material.

Unfortunately, available samples were not suitable to perform qPCR assays. To overcome this issue and to strengthen the data on the enhanced lysosomal biogenesis after treatment, we have performed immunostaining for another lysosomal marker, CD63, which agrees with the LAMP1 staining data. Please refer to figure 2N.

2. Following on patient data, in addition to my previous request, or as an alternative approach, the authors could use publicly available RNAseq data from BC patients treated with aromatase + Cdk4/6 inhibitors to investigate whether enhanced expression of lysosomal genes and lysosomal-associated pathways is a general feature. As an example please see doi:10.1038/s41523-024-00625-7 for data availability.

We requested access to the dataset as suggested and followed up over a two-month period. We contacted the owners of the dataset via email and submitted a request through the EGA website. Unfortunately, we did not receive a response initially, and access was ultimately denied. As such, we are unable to perform the requested analysis.

3. There is limited information regarding the BC biopsies used in the study. At least the authors should mention whether the post-Cdk4/6 biopsy was collected at the time of clinical response or at progression. We have now added the following statement in the M&M section: "The procedure was performed before starting the treatment and at the time of the best clinical response, respectively 6.9, 11.8 and 7.6 months after starting treatment. At last follow up, the three patients were disease-free."

4. While the phenotypic consequences of combining Cdk4/6i and lysosomotropic agents are well characterized in vitro, this is not the case in vivo (Figures 3J/K & 4J/K) where the efficacy is exclusively assessed by measuring tumour size and mass. At least the authors should evaluate whether in vivo the response is cytostatic or cytotoxic by performing immunostaining/immunofluorescence of apoptotic markers in tumour sections following the different treatments.

For our experimental setting, apoptosis markers are unlikely to yield results for two reasons: first, the cells most likely died through an apoptosis-independent mechanism (please also see new figure EV 2G; where we show that death is only partially dependent on caspase activation); and second, even if apoptosis occurred, the tumors were collected at a late time point, by which any cell death markers would have likely disappeared. Therefore, we examined the tumors for necrotic areas. However, the differences were not clear and difficult to compare, likely because the tumors were collected at a late stage after treatment, when immune cell activity and cell turnover could have influenced the extent of necrosis and the detection of any type of cell death.

Referee #3:

The manuscript by Nehme and collaborators studies the connection between CDK4/6 inhibitors and lysosomes in breast cancer cells, investigating the role of cellular senescence in this context. The authors describe that treatment with Abemaciclib causes changes in lysosome mass and structure in breast cancer cells, and show that these lysosomal alterations lead to increased sensitivity to lysosomotropic agents. The manuscript addresses an interesting question with potential translational implications. However, in its present form, the advancement relative to the current knowledge seems limited. The general link of cellular senescence to the lysosomal compartment is well established. Also, Abemaciclib (and other CDK4/6i) have previously been connected with lysosome function and senescence in several tumor cell lines, including some used in this study (Martinez-Carreres 2019 PMID 31395606, Li 2023 PMID 37993606, Rovira 2022 PMID 36087066), and the sensitivity of senescent cells (Curnock 2023 PMID 36970883) or CDK4/6i-treated cells (Llanos 2019 PMID 30692638) to lysosomotropic agents has been reported. A potentially relevant contribution of the study would be to establish whether or not this effect of CDK4/6i on lysosomes is necessarily associated to cellular senescence. Unfortunately, the current data about this question is somewhat preliminary and more work is needed to support the conclusions.

We would like to thank the reviewer for their thoughtful comments and valuable feedback. We hope our responses and discussions have adequately addressed their concerns and contributed to a clearer presentation of our research.

Major points

1. The potential link to senescence is an important issue of this study. Previous studies have reported that Abema, and other CDK4/6i, can induce a full senescent phenotype and senolytic sensitivity in tumor cell lines (Rovira 2022, Martinez-Carreres 2019, Li 2023 and others). In this manuscript, the authors conclude that the effect of CDK4/6i on lysosomes can be dissociated from full senescence, based on the apparent lack of cell-cycle arrest in MDA-MB-231 cells. This is a critical point that should be characterized in more detail. Late time points in the cell count experiment (Figure ED3a) seem to indicate reduced

proliferation with Abema and the images of the clonogenic assays (Figure ED3b) are not very informative. More sensitive assays, such as EdU incorporation, should be used to assess proliferation, and markers of senescence arrest, like p21, should also be monitored.

In the paper titled "TFEB-dependent lysosome biogenesis is required for senescence" (Curnock et al 2023) the primary focus was on the role of TFEB in the establishment of senescence, particularly in normal cells. This role was highlighted during the initial stages of senescence induction. The study noted that knocking down TFEB either at an early stage or prior to the induction of senescence significantly reduced cell viability, suggesting that the transcriptional activity of TFEB is crucial for maintaining cell survival in the early phases of senescence induction.

There are no studies to date showing that abemaciclib treatment sensitizes cells to typical senolytics, and this is what we demonstrated at the beginning of our study. While it is known that abemaciclib, like other "senescence inducers," can alter lysosomal content, we are the first to document the use of abemaciclib followed by lysosomotropic agents to effectively kill cancer cells. Our findings indicate that the effect is not solely dependent on the quantity of lysosomes but also on the structural changes in lysosomes induced by abemaciclib—changes that are not as pronounced with other treatments.

Moreover, we highlighted that cancer cells are much more vulnerable to this combined sequential therapy than normal cells. This selectivity emphasizes the potential of our strategy for precise cancer cell targeting, distinguishing our findings from previous studies.

In the two studies titled "Lysosomal trapping of palbociclib and its functional implications" (Llanos et al 2019) and "Increased lysosomal biomass is responsible for the resistance of triple-negative breast cancers to CDK4/6 inhibition" (Fassl et al 2020), the focus was not on cell death. Instead, these studies utilized various lysosomotropic agents to deacidify lysosomes and combat resistance to CDK4/6 inhibitors. It is important to note that not all lysosomotropic agents are identical; they do not produce uniform responses in cells, even though their target is the same—lysosomes. For instance, our research demonstrated that bafilomycin, which reduces lysosomal acidity, did not impact cell viability.

Furthermore, both studies applied a co-treatment approach from the beginning, treating the cells with lysosomotropic agents alongside CDK4/6 inhibitors. In contrast, our approach involved a sequential treatment where lysosomotropic agents were introduced only after the senescence-like phenotype began to manifest.

In the study titled "CDK4 regulates lysosomal function and mTORC1 activation to promote cancer cell survival" (Martínez-Carreres et al 2019), the researchers used a combination therapy from the beginning of the treatment. Specifically, cells were cultured for 8 days with a combination of an AMPK activator and a CDK4 inhibitor, leading to approximately 30% of cells becoming Annexin V-positive. This approach is fundamentally different from ours and, therefore, cannot be considered a senolytic therapy. Moreover, the cell death observed in their study was autophagic and linked to impaired lysosomal function, which contrasts with our findings. In our study, we demonstrated that bafilomycin A1, which deacidifies lysosomes and reduces their functionality, was unable to kill cells pre-exposed to abemaciclib. This further highlights the distinct mechanisms at play in our research compared to the approach taken in their study.

For additional proliferation and senescence markers please check the answers below.

As pointed out by the authors, it is possible that the treatment regimens used might account for the contrasting results on proliferation/senescence from different studies. It would be interesting if the authors could address this question directly in their experimental settings. For instance, what is the impact of

Abema on proliferation in MDA-MB-231 cells maintained in drug-containing medium compared to the current conditions where drug is withdrawn before the functional assays?

The proliferation of MDA-MB-231 cells, both during exposure to abemaciclib and after its removal, was evaluated using a colony formation assay and EdU staining. Our results demonstrate that in the presence of abemaciclib, MDA-MB-231 cells exhibit significantly reduced proliferative capacity. However, upon the removal of abemaciclib from the media, the cells promptly resume their cell cycle. Please refer to Figures EV4A and B.

In this context, it would be important to characterize better the senescence-like phenotype induced by Abema in MCF7 cells (Figure 1). Additional senescence markers should be monitored by QPCR, IF or WB. It would also be interesting to know if they display a specific SASP profile as previously described by the authors in Abema-treated non-transformed fibroblasts (Wang 2022 PMID 34985783).

If the difference in cell-cycle arrest is confirmed, it would be interesting to know if this is a TNBC-specific phenotype. Do the other HR+ cell lines show a full senescent phenotype, similarly to MCF7?

We evaluated proliferation in MCF-7 cells using an EdU assay and observed a significant reduction upon treatment with abemaciclib. Analysis of senescence markers revealed a marked upregulation of p21, while p16 remained undetectable. Notably, consistent with previous findings, there was no upregulation of the inflammatory SASP. However, genes regulated by p53 exhibited increased expression. Please refer to Figures 1C and EV4 A for detailed data supporting these observations. All the HR+ cells showed similar senescence responses.

2. The authors conclude that Abema-treated BJ normal fibroblasts are not sensitive to Salinomycin based on the data from a co-culture experiment with MCF7 cells (Figure ED2). How do they interpret this result considering their own report (Wang 2022) that Abema renders BJ fibroblasts senescent with gene expression enrichment in lysosomal-associated pathways? The potential difference between cancer and normal cells is an important question that deserves more attention. It would be more informative to perform assays in BJ cells alone, analogous to the ones shown for breast cancer cells (expression of lysosomal markers, sensitivity to senolytics and lysosomotropic drugs).

We evaluated the sensitivity of two non-cancerous cell lines, BJ and IMR90, to lysosomotropic agent-induced cell death following abemaciclib treatment, separately from cancer cells. No significant difference in sensitivity was observed. Please refer to Figure EV3 A for the results. While this observation is intriguing, elucidating the mechanisms underlying the greater sensitivity of cancer cells to the drug compared to normal cells is beyond the scope of this study. However, this remains an important and compelling question for future research.

3. Figure 5 shows that vacuolin increases the lysosomal compartment and induces vulnerability to lysosomotropic drugs. Vacuolization has been associated with senescence. Did vacuolin treatment have any effect on senescence markers?

We assessed senescence markers following treatment with vacuolin. There was no change in cell proliferation, and no increase in SA- β -gal staining was observed, indicating that vacuolin treatment does not induce senescence. Please refer to figures EV5 A and B.

4. The manuscript would be improved if the authors could elaborate more on the potential mechanism. The results with the Rb-negative BT549 cell line (Fig ED 3) suggest that the impact of CDK4/6i on

lysosomes is linked to the on-target effect on the cell-cycle machinery. In addition, the data in Figure 5 suggest a critical role of vacuolization. It would be interesting to discuss whether these two processes might be connected. Interestingly, other reports have suggested that the effect of CDK4/6i on lysosomes involves mTOR signaling (Martinez-Carreres 2019). This potential mechanism should also be discussed and, ideally, tested experimentally with pharmacological perturbation of this pathway.

Concerning the Effect of RB1 Knockdown on lysosomal biogenesis, we preformed knocked down experiments. We observed a decrease in lysosomal biogenesis, which correlated with reduced cell sensitivity to death induced by lysosomotropic agents after treatment with abemaciclib. This suggests that mechanisms downstream of RB suppression may inhibit lysosomal biogenesis at the RNA level. One potential explanation is the role of CDK1, which has been implicated in the suppression of TFEB activity, suggesting that continued cell cycling may inhibit TFEB and thereby reduce lysosomal biogenesis. Please refer to figure EV 2 H and I.

In the case of resistance to CDK4/6 Inhibition in BT549 cells, we examined whether abemaciclib can effectively inhibit CDK4/6 activity in this cell line. Using a CDK4/6 activity probe, we found that BT549 cells do not exhibit any response to CDK4/6 inhibition, and CDK4/6 activity remains unaffected. This suggests the presence of upstream mechanisms of resistance. Potential explanations for this resistance could include multidrug resistance mechanisms, restricted accessibility to CDK4/6, or other yet unidentified pathways. Please refer to figure EV 4I.

Please check the answers for cited articles.

5. Several relevant articles currently not mentioned in the manuscript should be cited and discussed: Llanos 2019 PMID 30692638, Martinez-Carreres 2019 PMID 31395606, Rovira 2022 PMID 36087066, Li 2023 PMID 37993606, Curnock 2023 PMID 36970883 among others.

These articles have now been cited in the manuscript; please refer to the discussion section.

Minor points

1. The current display and description in the text of differential gene expression data from an independent study (Goel 2017 PMID 28813415) is somewhat misleading (Figures 1c, d, 2a, b and ED1a, b). It should be clearly stated in the main text that this is not original data from the current study. There is also a concern that the conditions used in both studies do not appear to be the same (1 μ M Abemaciclib for 8 days in the current manuscript versus 250 nM for 7 days in Goel 2017).

To further validate our observations, we checked various treatment concentrations and durations using data from online datasets. Please see Figure EV 2C, which demonstrates the upregulation of lysosomal gene expression across a range of conditions. We specify that publicly-available RNA sequencing datasets were utilized, with all corresponding accession numbers cited in the figures.

2. In the lysosome stainings in Figure 2d and ED 1c, d, please show quantifications and similar magnification examples for all the markers.

In Figure 2D, acridine orange staining is shown as a representative image, without quantification, as a more accurate analysis using flow cytometry is presented in Figure 2G. LAMP1 and Cytofix staining were also analyzed via flow cytometry, with their quantifications provided in Figures 2H and 2I. Additionally, in Figure EV2E, LAMP1 imaging was used to assess both structural and quantitative changes in lysosomes. Please note that the positions of the panels have been rearranged following the revision.

3. In the spheroid assay (Figure 3j), Salinomycin alone causes significant cell death but this effect is not obvious in the tumorigenesis assay (Figure 3k). How do the authors interpret these results? Also, Abema alone seems to have a negligible effect in the tumorigenesis assays (Figure 3k). Wouldn't one expect to see a reduction in tumor growth consistent with senescence induction?

These results can be attributed to different factors. First, the assay does not account for the irregular size and shape of the organoids, making it difficult to normalize results. Additionally, despite the 3D structure of the organoids, the assay measures cell death on a 2D plane. As a result, any change in surface area—such as the reduction seen with abemaciclib and salinomycin treatment or the enlargement observed with salinomycin alone—can affect the cell death signal. This occurs because the measurement is not normalized to the actual number of cells. Despite these limitations, the assay still demonstrates the effectiveness of the dual treatment in inducing cell death. Our treatment resulted in tumor volume stabilization during abemaciclib administration. However, once abemaciclib was removed, the tumor resumed growth, indicating that MCF7 cells were either not fully arrested or not all had entered senescence. This may be due to the lower dose and shorter treatment duration used in our regimen compared to other studies.

4. I recommend using the standard name MDA-MB-231 (Cellosaurus CVCL_0062) to designate the cell line called MDA-231 in the manuscript.

We have adjusted the name to the standardized nomenclature (MDA-MB-231).

5. Please clarify if the indicated number of samples are biological or technical replicates.

We stated the number of independent experiments conducted.

Dear Dr Demaria,

Thank you for submitting your revised manuscript (EMBOJ-2024-118394R) to The EMBO Journal, as well for your patience with our response. Your amended study was sent back to the three referees for their scientific re-evaluation, and we have received detailed comments from all of them, which I enclose below. As you will see, the experts state that the work has been substantially enhanced by the revisions and they are now broadly in favour of publication, pending minor revision.

Thus, we are pleased to inform you that your manuscript has been accepted in principle for publication in The EMBO Journal.

Please carefully consider the remaining minor points raised by reviewer #3 regarding statements made on results and overall wording of the manuscript and adjust the text where appropriate. Please also revisit and complement the methods annotation as required.

In addition, we now need you to take care of a number of issues related to formatting and data presentation as detailed below, which should be addressed at re-submission.

Please contact me at any time if you have additional questions related to below points.

As you might remember from previous experience, every paper at the EMBO Journal now includes a 'Synopsis', displayed on the html and freely accessible to all readers. The synopsis includes a 'model' figure as well as 2-5 one-short-sentence bullet points that summarize the article. I would appreciate if you could provide this figure and the bullet points.

Thank you for giving us the chance to consider your manuscript for The EMBO Journal. I look forward to your final revision.

Again, please contact me at any time if you need any help or have further questions.

Best regards,

Daniel Klimmeck

>> Please add up to five keywords to your study.

>> Author Contributions: Remove the author contributions information from the manuscript text. Note that CRediT has replaced the traditional author contributions section as of now because it offers a systematic machine-readable author contributions format that allows for more effective research assessment. and use the free text boxes beneath each contributing author's name to add specific details on the author's contribution.

More information is available in our guide to authors.
<https://www.embopress.org/page/journal/14602075/authorguide>

>> Adjust the title of the 'Competing Interests' section to 'Disclosure and Competing Interests Statement' and move after

Acknowledgements.

>> Correct order of manuscript sections: Abstract / Keywords / Introduction / Results / Discussion / Methods / Data Availability / Acknowledgements / Disclosure and competing interests statement // References / Figure legends / Tables and their legends / Expanded View Figure legends

>> The heading "main text" should be removed and replaced by "Introduction".

>> "Materials and Methods" should be renamed "Methods".

>> The table on pp 14-5 should be named "Table 1" and moved after the main figure legends: a legend should be added and a callout.

>> The heading "Declarations" should be removed and the "Ethics Approval" should be moved to the Methods.

>> 'Funding' section should be renamed to "Acknowledgments"; project numbers included in our online system should also be added to the manuscript text.

>> References: please adjust reference format to EMBO Journal format, 10 authors et al. .

>> Figures in separate files: the EV figures should also be uploaded as individual, high-resolution figure files.

>> Data availability section: include a 'Data availability section' stating: 'No data amenable to large-scale repository deposition were generated in this study.' .

>> Add a separate 'Statistical analysis' section, detailing the algorithms and statistical tests applied.

>> Add a Reagents and Tools table to the Methods section, as a separate file using the existing template in the Guide For Authors, listing key reagents, experimental models, software and relevant equipment.

>> Source data: source data should be uploaded as one (zipped) file per figure.

>> Consider additional changes and comments from our production team as indicated below:

- Figure legends:

1. Please note that the exact p values are not provided in the legends of figures 1A, C; 2C, G, H, I, L, M, N; 3A, D, G, H, I, J, K; 4B, E, F, G, I, J, K; 5A, C, F, G, I; EV1 A, EV2 D-I; EV3 C, EV4 B, C, D, H, I; 5B
2. Please indicate the statistical test used for data analysis in the legends of figures 2B, 4K, EV2 A.
3. Please note that information related to n is missing in the legends of figures 2B, EV4 B, C.
4. Please note that the error bars are not defined in the legends of figures 2C, G, H, I, L; 4K.
5. Please note that the scale bar needs to be defined for figures 2M, N; 5H.

Referee #1:

Authors has provided new data providing mechanistic insights linking CDK4/6 inhibition to vulnerability to lysosomotropic agents, such as L-leucyl-L-leucine methyl ester (LLOMe) and salinomycin. This is now in my opinion a nice paper with great translational potential.

Referee #2:

The authors have addressed most of my requests. It is unfortunate that access to the EGA repository was denied. As an alternative the authors could request expression information/fold change data restricted to their dataset of interest (lysosomal genes and lysosomal-associated pathways). I leave this possibility at both the author's and editor's discretion.

Referee #3:

In the revised version, the authors have included new experimental data and changes in the manuscript that addressed several of the points that I raised. However, in my opinion, there are still some issues that deserve attention.

1. As mentioned in my comments to the first version, the potential link to senescence is an important question in this study. The authors have included new data (EdU incorporation, QPCR of senescence markers) that improves the characterization of the phenotype of Abemaciclib-treated MCF7 cells. However, despite the claim in the rebuttal that "All the HR+ cells showed similar senescence responses" (point 1 of rebuttal), the current data (Figs 2J-L, 3G-I, EV2F) shows similar lysosome-related changes and sensitivity to salinomycin, but other lysosome-independent senescence markers were not studied. So, it is difficult to know how general is the senescence induction by Abemaciclib in HR+ cells.

On the other hand, the tumorigenesis assays show a clear combined effect of Abemaciclib and Salinomycin, even though "MCF7 cells were either not fully arrested or not all had entered senescence" (point 3 of rebuttal). If the authors' interpretation is correct, the impact of Abemaciclib and Salinomycin in this setting does not seem to require a full senescent phenotype (i.e., including arrest), in line with some of the in vitro data.

Although there are still some gaps that make difficult to reach conclusions, the current data seems to suggest that the combined effect of K4/6i and lysosomotropic agents is associated with senescence-related lysosomal changes but it does not necessarily require a full senescent phenotype. I believe this is an important issue in this work and it should be commented upon in the Discussion.

2. The statement in the section caption "Abemaciclib induces a senescence-like phenotype in TNBC" (page 6) is not fully supported by the current data. Cell-cycle arrest was only observed under some conditions, and additional lysosome-independent senescence markers were not tested. In the context of this study, the expression "senescence-like" can be misleading. I would recommend to distinguish lysosome-related phenotypes, which seem to be present in all cases, from other senescence-related lysosome-independent markers, like cell cycle arrest or SASP

3. Similarly, the statement "sequential therapy can also be applied to TNBCs" (page 7) should be rephrased as only one of the two TNBC cell lines tested showed a positive response. A more balanced description of the potential impact in TNBC is given in the Discussion (page 9).

4. Regarding the cell death data in spheroids (Figure 3J, 4I), the authors' comments in the rebuttal (point 3) raise doubts about the reliability of the assay. Taking into account the limitations pointed out by the authors it is questionable that this method can be used to accurately quantify cell death in this particular setting.

5. The TFEB-GFP vector used in Figure EV2D of the new version is not described in Material and Methods.

Editorial:

>> Please add up to five keywords to your study.

AUTHORS: Added.

>> Author Contributions: Remove the author contributions information from the manuscript text. Note that CRediT has replaced the traditional author contributions section as of now because it offers a systematic machine-readable author contributions format that allows for more effective research assessment. and use the free text boxes beneath each contributing author's name to add specific details on the author's contribution.

This has now been removed from the main text.

>> Adjust the title of the 'Competing Interests' section to 'Disclosure and Competing Interests Statement' and move after Acknowledgements.

Done.

>> Correct order of manuscript sections: Abstract / Keywords / Introduction / Results / Discussion / Methods / Data Availability / Acknowledgements / Disclosure and competing interests statement / / References / Figure legends / Tables and their legends / Expanded View Figure legends

Done.

>> The heading "main text" should be removed and replaced by "Introduction".

Done, and the "Results" section was added.

>> "Materials and Methods" should be renamed "Methods".

Done.

>> The table on pp 14-5 should be named "Table 1" and moved after the main figure legends: a legend should be added and a callout.

Done.

>> The heading "Declarations" should be removed and the "Ethics Approval" should be moved to the Methods.

Done.

>> 'Funding' section should be renamed to "Acknowledgments"; project numbers included in our online system should also be added to the manuscript text.

Modified.

>> References: please adjust reference format to EMBO Journal format, 10 authors et al. .

Adjusted.

>> Figures in separate files: the EV figures should also be uploaded as individual, high-resolution figure files.

Done.

>> Data availability section: include a 'Data availability section' stating: 'No data amenable to large-scale repository deposition were generated in this study.' .

Added in the text.

>> Add a separate 'Statistical analysis' section, detailing the algorithms and statistical tests applied.

Added.

>> Add a Reagents and Tools table to the Methods section, as a separate file using the existing template in the Guide For Authors, listing key reagents, experimental models, software and relevant equipment.

Added.

>> Source data: source data should be uploaded as one (zipped) file per figure.

Added.

>> Consider additional changes and comments from our production team as indicated below:

- Figure legends:

1. Please note that the exact p values are not provided in the legends of figures 1A, C; 2C, G, H, I, L, M, N; 3A, D, G, H, I, J, K; 4B, E, F, G, I, J, K; 5A, C, F, G, I; EV1 A, EV2 D-I; EV3 C, EV4 B, C, D, H, I; 5B

Added in separate figure legends.

2. Please indicate the statistical test used for data analysis in the legends of figures 2B, 4K, EV2A.

Added.

3. Please note that information related to n is missing in the legends of figures 2B, EV4 B, C.

Added.

Referee #1:

Authors has provided new data providing mechanistic insights linking CDK4/6 inhibition to vulnerability to lysosomotropic agents, such as L-leucyl-L-leucine methyl ester (LLOMe) and salinomycin. This is now in my opinion a nice paper with great translational potential.

Thank you, we appreciate your support and your help in improving the manuscript.

Referee #2:

The authors have addressed most of my requests. It is unfortunate that access to the EGA repository was denied. As an alternative the authors could request expression information/fold change data restricted to their dataset of interest (lysosomal genes and lysosomal-associated pathways). I leave this possibility at both the author's and editor's discretion.

Unfortunately we were unable to communicate with the authors.

Referee #3:

In the revised version, the authors have included new experimental data and changes in the manuscript that addressed several of the points that I raised. However, in my opinion, there are still some issues that deserve attention.

1. As mentioned in my comments to the first version, the potential link to senescence is an important question in this study. The authors have included new data (EdU incorporation, QPCR of senescence markers) that improves the characterization of the phenotype of Abemaciclib-treated MCF7 cells. However, despite the claim in the rebuttal that "All the HR+ cells showed similar senescence responses" (point 1 of rebuttal), the current data (Figs 2J-L, 3G-I, EV2F) shows similar lysosome-related changes and sensitivity to salinomycin, but other lysosome-independent senescence markers were

not studied. So, it is difficult to know how general is the senescence induction by Abemaciclib in HR+ cells.

On the other hand, the tumorigenesis assays show a clear combined effect of Abemaciclib and Salinomycin, even though "MCF7 cells were either not fully arrested or not all had entered senescence" (point 3 of rebuttal). If the authors' interpretation is correct, the impact of Abemaciclib and Salinomycin in this setting does not seem to require a full senescent phenotype (i.e., including arrest), in line with some of the in vitro data.

Although there are still some gaps that make difficult to reach conclusions, the current data seems to suggest that the combined effect of K4/6i and lysosomotropic agents is associated with senescence-related lysosomal changes but it does not necessarily require a full senescent phenotype. I believe this is an important issue in this work and it should be commented upon in the Discussion.

We appreciate the reviewer's insightful comments and would like to emphasize that the primary focus of our study was to identify a target in cells treated with abemaciclib, which is known to induce a senescence-like state in many cell types. The study was not specifically designed to determine whether a fully senescent state is required for the vulnerability induced by abemaciclib. Our findings indeed indicate that lysosomal changes associated with senescence likely represent the main target. However, we recognize that a cell cycle arrest, even if transient, may contribute to this process, as it could influence the lysosomal phenotype of "senescent-like" cells.

2. The statement in the section caption "Abemaciclib induces a senescence-like phenotype in TNBC" (page 6) is not fully supported by the current data. Cell-cycle arrest was only observed under some conditions, and additional lysosome-independent senescence markers were not tested. In the context of this study, the expression "senescence-like" can be misleading. I would recommend to distinguish lysosome-related phenotypes, which seem to be present in all cases, from other senescence-related lysosome-independent markers, like cell cycle arrest or SASP

We would like to thank the reviewer for their comments and acknowledge the concern regarding the wording. In describing our data, in many cases we deliberately avoided using the term "senescence" which reflects a fully senescent state. Instead, we employed the term "senescence-like," which we believe more accurately describes a condition where cells exhibit hallmark features of senescence but may not fully meet the classical criteria or exhibit irreversibility. This distinction is important, as senescence-like cells can sometimes resume proliferation if the inducing stressors are removed or reversed. Consequently, this phenotype may represent a transient cellular response to

stress rather than a permanent state. Accordingly, a transient cell cycle arrest accompanied by an increase in β -galactosidase activity can be characterized as a senescence-like phenotype. If necessary, we are willing to elaborate further on this distinction and its implications in the discussion section.

3. Similarly, the statement "sequential therapy can also be applied to TNBCs" (page 7) should be rephrased as only one of the two TNBC cell lines tested showed a positive response. A more balanced description of the potential impact in TNBC is given in the Discussion (page 9).

We have now rephrased the last sentence of the section as: "These data suggest that sequential therapy might be applied to triple-negative breast cancer, provided that abemaciclib inhibits CDK4/6 activity".

4. Regarding the cell death data in spheroids (Figure 3J, 4I), the authors' comments in the rebuttal (point 3) raise doubts about the reliability of the assay. Taking into account the limitations pointed out by the authors it is questionable that this method can be used to accurately quantify cell death in this particular setting.

We understand the concerns raised and appreciate the feedback. However, it is important to note that we did not rely on this method for precise quantification of cell death. Accurate quantification was performed using mCherry-labeled cells. This method was utilized to provide a relative quantification of cell death in 3D tumor models. While we acknowledge its limitations, as with other methods, it remains a valuable addition to our manuscript when complemented by more accurate quantitative approaches, such as the use of labeled H2B-mCherry cells in combination with CellTox Green in live-cell imaging.

5. The TFEB-GFP vector used in Figure EV2D of the new version is not described in Material and Methods.

It has been included in the Methods section of the manuscript.

Dear Dr Demaria,

Thank you for submitting the revised version of your manuscript. I have now evaluated your amended manuscript and concluded that the remaining minor concerns have been sufficiently addressed.

I am thus pleased to inform you that your manuscript has been accepted for publication in the EMBO Journal.

On a different note, I would like to alert you that EMBO Press offers a format for a video-synopsis of work published with us, which essentially is a short, author-generated film explaining the core findings in hand drawings, and, as we believe, can be very useful to increase visibility of the work. Please see the following link for representative examples and their integration into the article web page:

<https://www.embopress.org/doi/full/10.15252/emj.2019103932>

Finally, we have noted that the submitted version of your article is also posted on the preprint platform bioRxiv. We would appreciate if you could alert bioRxiv on the acceptance of this manuscript at The EMBO Journal in order to allow for an update of the entry status. Thank you in advance!

Best regards,

Daniel Klimmeck

Daniel Klimmeck, PhD
Senior Editor
The EMBO Journal
EMBO
Postfach 1022-40
Meyerohofstrasse 1
D-69117 Heidelberg
contact@embojournal.org